# Gadd45α modulates aversive learning through post-transcriptional regulation of memory-related mRNAs

Alejandro Aparisi Rey[1], Emil Karaulanov[2], Salim Sharopov[3], Khelifa Arab[2], Andrea Schäfer[2], Mathias Gierl[2], Stephan Guggenhuber[1], Caroline Brandes[1], Luigi Pennella[1], Wolfram H Gruhn[2], Ruth Jelinek[1], Christina Maul[1], Andrea Conrad[1], Werner Kilb[3], Heiko J Luhmann[3], Christof Niehrs[2,4,*] & Beat Lutz[1,**]

## Abstract

Learning is essential for survival and is controlled by complex molecular mechanisms including regulation of newly synthesized mRNAs that are required to modify synaptic functions. Despite the well-known role of RNA-binding proteins (RBPs) in mRNA functionality, their detailed regulation during memory consolidation is poorly understood. This study focuses on the brain function of the RBP Gadd45α (growth arrest and DNA damage-inducible protein 45 alpha, encoded by the *Gadd45a* gene). Here, we find that hippocampal memory and long-term potentiation are strongly impaired in *Gadd45a*-deficient mice, a phenotype accompanied by reduced levels of memory-related mRNAs. The majority of the Gadd45α-regulated transcripts show unusually long 3′ untranslated regions (3′UTRs) that are destabilized in *Gadd45a*-deficient mice via a transcription-independent mechanism, leading to reduced levels of the corresponding proteins in synaptosomes. Moreover, Gadd45α can bind specifically to these memory-related mRNAs. Our study reveals a new function for extended 3′UTRs in memory consolidation and identifies Gadd45α as a novel regulator of mRNA stability.

**Keywords** *Gadd45a*; *Grin2a*; memory; RNA stability
**Subject Categories** Neuroscience; RNA Biology

## Introduction

Memory consolidation requires the synthesis of new transcripts and proteins. Thus, mechanisms controlling these processes are central for understanding memory formation. Among these processes, activity-dependent post-transcriptional regulation at the mRNA level has profound consequences on RNA structure and subcellular localization [1]. This, in turn, affects the efficiency of translation of memory-related genes, such as *brain-derived neurotrophic factor* (*bdnf*) [2] and the subunit epsilon 1 of the ionotropic glutamate NMDA receptor (*Grin2a*) [3]. In this context, the 3′ untranslated region (3′UTR) has been classically considered as a central hub for post-transcriptional modifications, including RNA stabilization and/ or translational facilitation or inhibition. These modifications are frequently modulated by RNA-binding proteins (RBPs), although little is known about the exact underlying mechanism. Thus, the discovery of new RBPs is fundamental to unravel the regulation of mRNA functionality.

The growth arrest and DNA damage-inducible protein 45 (Gadd45) family comprises three structurally related proteins: Gadd45α, Gadd45β, and Gadd45γ. They are small acidic proteins, localized in the nucleus and cytoplasm, and play essential roles in stress response, cell cycle regulation [4], modulation of mitogen-activated protein kinases (MAPKs) [5], and the regulation of DNA demethylation [6–8]. Interestingly, Gadd45α has recently been described as a member of the L7a class of RNA-binding proteins, which is present in nuclear speckles [9] that are sites of transcription and post-transcriptional processing. Moreover, Gadd45α is widely expressed in the mouse brain, including cerebral cortex and hippocampus [10], fundamental regions for memory formation. Thus, Gadd45α has the potential to control brain functions by altering the availability of functional mRNA. In fact, two recent reports failed to provide a consistent view on the role of Gadd45β in the context of memory formation. While Leach *et al* [11] reported that *Gadd45b*-deficient mice exhibit deficient hippocampus-dependent memory, Sultan *et al* [12] revealed an increase in memory formation and synaptic plasticity on the same mutant mice. Here, we investigated *Gadd45a*-deficient mice (*Gadd45a*-KO), in which exons 1–3 were replaced by a phosphoglycerate

1 Institute of Physiological Chemistry, University Medical Center of the Johannes Gutenberg University Mainz, Mainz, Germany
2 Institute of Molecular Biology, Mainz, Germany
3 Institute of Physiology, University Medical Center of the Johannes Gutenberg University Mainz, Mainz, Germany
4 Division of Molecular Embryology, DKFZ-ZMBH Alliance, Deutsches Krebsforschungszentrum (DKFZ), Heidelberg, Germany
*Corresponding author. Tel: +49 6131 39 21400; E-mail: c.niehrs@imb-mainz.de
**Corresponding author. Tel: +49 6131 39 25912; E-mail: beat.lutz@uni-mainz.de

kinase–neomycin cassette [13]. *Gadd45a*-KO mice are viable and fertile, indicating that Gadd45α is not fundamental for mouse development. However, they exhibit a low frequency of exencephaly (8% of pups) and are prone to develop genomic instability and cancer [13]. Interestingly, several of these phenotypes are shared with p53-deficient mice [13]. Despite the extensive literature in the field of cancer research [14,15], a role for Gadd45α in the central nervous system has so far not been detailed.

In this study, we analyzed *Gadd45a*-deficient as well as *Gadd45a*-overexpressing mice in learning and memory paradigms, applying a combination of behavioral, molecular, electrophysiological, and transcriptomic techniques. We found that Gadd45α is essential for the consolidation of aversive memories and that it affects the stability of transcripts with extended 3′UTRs being involved in memory and synaptic plasticity. In order to detail the physiological importance of this process, we also studied the transcriptional consequences during memory consolidation and the capacity of Gadd45α to bind its potential mRNA targets.

# Results

### Gadd45α deficiency leads to impaired memory consolidation

To identify neural functions regulated by Gadd45α, we performed a comprehensive behavioral characterization of *Gadd45a*-KO mice. A large number of behavioral parameters were found to be unaltered compared to wild-type littermates (*Gadd45a*-WT), including locomotion and exploration (Fig EV1A), anxiety (Fig EV1B and C), and depressive-like behavior (Fig EV1D). In addition, the ability to recognize familiar objects in an object recognition test (a non-aversive, non-emotional task) was not altered in mutants (Fig EV1E). However, genotype-specific differences were observed in paradigms of hippocampus-dependent learning. In the passive avoidance test (PA; Fig 1A), which depends on intact functions of the hippocampus [16,17], *Gadd45a*-KO mice showed a significant decrease in the latency to enter the dark (shock-associated) compartment as compared to *Gadd45a*-WT (Fig 1B), when tested 24 h after the training ($t_{20} = 2.295$, $P < 0.05$) and after 7 days ($t_{20} = 2.568$, $P < 0.05$). This difference was not observed when animals were tested 1 h after the training, indicating that memory consolidation but not acquisition depends on Gadd45α function. To corroborate these findings, we performed another hippocampus-dependent paradigm in which a correct spatial navigation is required, the water cross-maze (WCM [18,19]) (Fig EV1F). This paradigm is an adapted version of the classical Morris water maze, which has been used extensively in the study of hippocampus-dependent learning [20]. In this experiment, the accuracy to locate a hidden platform (in order to escape) by the animals increased over consecutive sessions. Both *Gadd45a*-WT and *Gadd45a*-KO mice significantly increased the accuracy between the first two sessions (acquisition phase) on day 1 (session factor: $F_{1,19} = 10.93$, $P < 0.01$). However, in sessions 3 and 5 *Gadd45a*-KO mice showed a significantly lower accuracy ($t_{19} = 3.214$, $P < 0.01$), confirming that the consolidation but not the acquisition phase was affected by Gadd45α deficiency. Consistently, this impaired consolidation was also observed when the location of the platform was changed and animals had to relearn the task (reversal learning; Fig EV1F). In a third aversive learning task,

cued fear conditioning, *Gadd45a*-KO mice also showed a decreased freezing response to a shock-associated tone as compared to wild-type controls (Fig EV1G). We conclude that Gadd45α is necessary for memory consolidation, specifically in aversive/stress-related learning paradigms.

Given a role of Gadd45α in hippocampal memory consolidation, we confirmed its reported expression in the hippocampus [10] by *in situ* hybridization (ISH). *Gadd45a* mRNA was evidently present in areas *cornu ammonis* 1–3 (CA1-CA3) and *dentate gyrus* (DG) of adult mice (Fig 1C; left panel). The signal localization suggests that *Gadd45a* mRNA is restricted to the stratum pyramidale of the hippocampus, a layer highly populated by the cell body of principal glutamatergic neurons. Negative controls, including sense probe on wild-type sections (Fig 1C; middle panel) and the antisense probe on sections from *Gadd45a*-KO mice (Fig 1C; right panel), gave weak background signals in the DG, indicating that the detection of *Gadd45a* mRNA in CA1-3 was specific. Quantification of *Gadd45a* mRNA by qPCR assays confirmed lower levels of *Gadd45a* mRNA in the hippocampus than of *Gadd45b* and *Gadd45g* (Fig EV1H). *Gadd45a* expression was very similar to other important regulators involved in learning and memory, such as *bdnf* and *reelin* (Fig EV1H). Unlike *Gadd45b*, *Gadd45a* was neither induced upon exposure to a new context (PA test without shock) nor induced upon exposure to the context plus foot shock (Fig 1D). Note that in both groups (context and context + shock), the absence of induction of *Gadd45a* mRNA was not caused by deficient neuronal activation, as the expression of *activity-regulated cytoskeleton* gene (*Arc*) was markedly increased.

### Gadd45a overexpression in the hippocampus improves memory consolidation

We next tested whether an increase in Gadd45α levels is able to enhance memory consolidation. We overexpressed Gadd45α specifically in the neuronal population that endogenously expresses Gadd45α (principal glutamatergic neurons of the hippocampus). To this end, we used an adeno-associated virus (AAV) containing *Gadd45a* fused to hemagglutinin (HA)-tag. We also introduced a transcriptional stop cassette (stop) flanked by loxP sites, to control *Gadd45a* expression (Fig 2A). This virus was injected into the dorsal hippocampus of heterozygous *Nex-Cre* mice (*Nex-Cre*$^{+/-}$), where Cre recombinase is expressed under the control of the regulatory regions of the *Nex* gene (i.e., only at principal glutamatergic neurons [21]). As a control, the same virus was injected into wild-type mice expressing no *Cre recombinase* (*Nex-Cre*$^{-/-}$). *Gadd45a* expression was validated by qPCR, showing a 12-fold overexpression of *Gadd45a* mRNA levels (Fig 2B). HA immunostaining revealed that Gadd45α-HA was overexpressed only in the expected areas (CA1-CA3) of the hippocampi of *Nex-Cre*$^{+/-}$ mice (Fig 2C). Gadd45α-HA was found mainly in the stratum pyramidale of CA1-CA3 (sp; neuronal bodies), with a weaker but significant expression in the stratum oriens and stratum radiatum (so and sr, respectively; i.e., in neuronal projections). In order to confirm that Gadd45α-HA expression was not restricted to the neuronal nuclei, we measured the presence of the protein in nuclear versus cytoplasmic protein fractions. The results confirmed that Gadd45α-HA is also present in the cytoplasm of hippocampal neurons (Fig 2D).

     

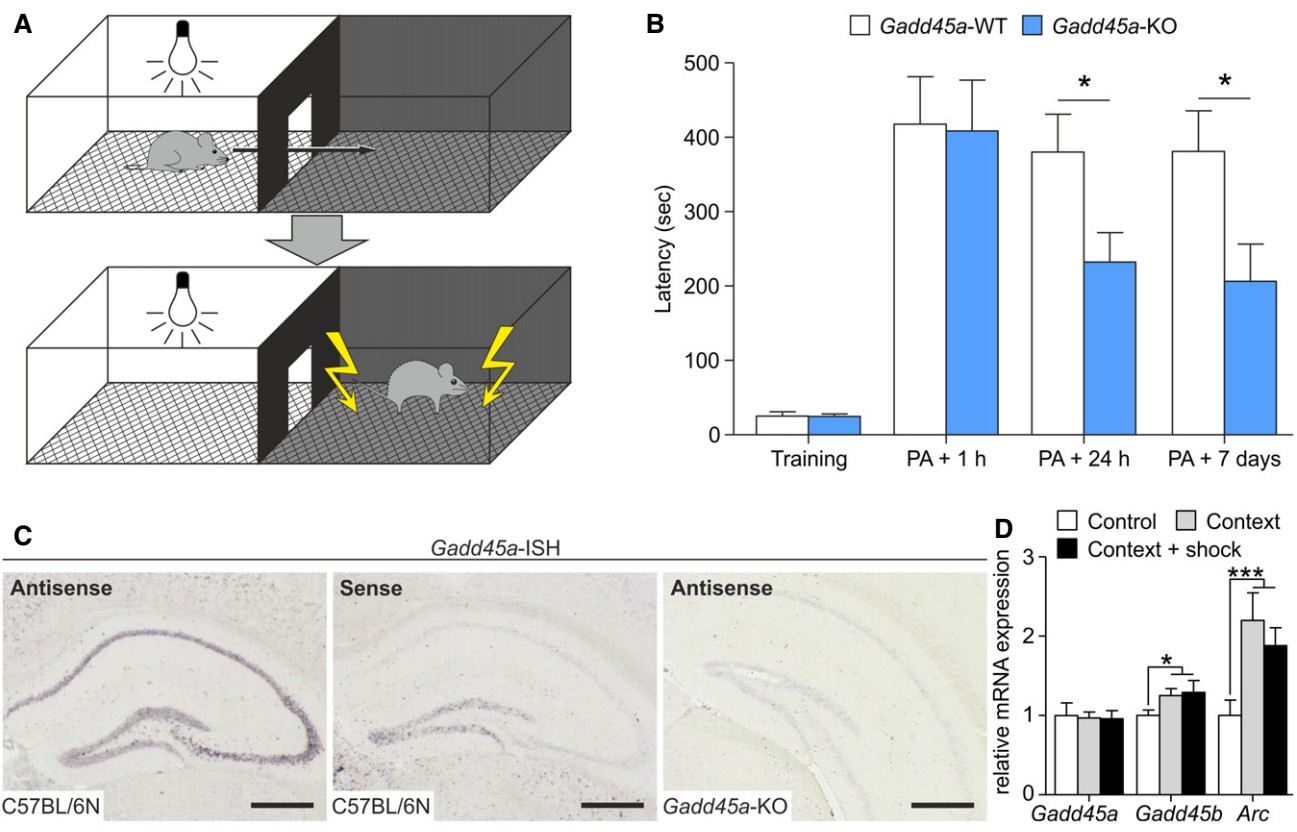

**Figure 1. Gadd45α controls hippocampus-dependent memories.**

A  Experimental apparatus of the passive avoidance (PA) test to assess context-dependent associative learning. In this paradigm, the animal receives a foot shock immediately after entering the dark compartment. This leads to hippocampal-dependent aversive learning, which can be assessed by the latency of subjects to enter the dark compartment in subsequent phases of the experiment.

B  Latency to enter dark compartment for *Gadd45a*-WT (white bars, *n* = 11) and *Gadd45a*-KO (blue bars, *n* = 11). A significant reduction was found in the *Gadd45a*-KO at 24 h and 7 days after the training. Values shown are mean ± SEM; two-way ANOVA and Bonferroni post hoc test: *$P$ < 0.05.

C  In situ hybridization to detect *Gadd45a* mRNA in adult wild-type hippocampus (*n* = 8 animals; antisense, left panel; sense, middle panel) and in *Gadd45a*-KO sections (*n* = 6 animals; right panel). Scale bars: 500 μm (black bar).

D  Experience-dependent changes in mRNA levels were evaluated under control conditions (home cage) and exposed to the PA (context and context + shock groups; *n* = 6 for all groups), respectively. Hippocampal *Gadd45a* mRNA levels remained unchanged, while *Gadd45b* was induced (context: $t_{10}$ = 2.427, context + shock: $t_{10}$ = 2.319). Concomitantly, mRNA levels of activity-regulated cytoskeleton protein (*Arc*), as a proxy of neuronal activation, were increased in mice exposed to the PA context ($t_{10}$ = 3.406) and context + shock ($t_{10}$ = 3.141). Values shown are mean ± SEM; one-way ANOVA and unpaired *t*-test: *$P$ < 0.05, ***$P$ < 0.001.

As predicted, these animals displayed increased memory performance in the PA paradigm, as latency to the dark compartment was prolonged 24 h ($t_{20}$ = 2.963, $P$ < 0.05) and 7 days ($t_{20}$ = 2.095, $P$ < 0.05) after the training (Fig 2E). Accordingly, the learning curve of these *Gadd45a*-overexpressing mice in the WCM was also improved as compared to WT controls (Fig EV2A). To exclude confounding factors, such as different pain sensitivity, we evaluated the pain threshold of both *Gadd45a*-KO and *Gadd45a*-overexpressing mice and their WT controls in two different assays. Both, the latency to the first reaction of discomfort in the hot plate test and the minimum intensity at which mice reacted to the foot shock (Fig EV2B and C), were unaltered in all the experimental groups. This demonstrated that the alterations in memory consolidation cannot be ascribed to differences in pain sensitivity. Taken together, the results of the overexpression model corroborated that Gadd45α promotes consolidation of hippocampus-dependent memories in tasks which include averseness.

## Gadd45α controls synaptic plasticity in hippocampus and amygdala

To elucidate the mechanisms underlying the Gadd45α-dependent memory consolidation, we evaluated synaptic plasticity both in *Gadd45a*-KO and in *Gadd45a*-overexpressing mice by inducing long-term potentiation (LTP) on hippocampal slices (Fig 3A). The LTP protocol was applied to hippocampi of 16 slices from five *Gadd45a*-WT mice and of 11 slices from five *Gadd45a*-KO mice. In all slices from *Gadd45a*-WT, a significant ($P$ < 0.05, Student's *t*-test) LTP could be induced. The field potential (FP) slope increased to 179 ± 8% (*n* = 16) and 165 ± 9% (*n* = 16) at 60 min and 90 min after LTP induction, respectively (Fig 3B). In contrast, in the hippocampus of *Gadd45a*-KO, the induction of LTP was impaired. In only 6 out of 11 slices, a significant LTP could be induced, which was a lower incidence ($P$ = 0.013, Fisher's exact test) than in *Gadd45a*-WT. The slope increased to 125 ± 8% (*n* = 11) and

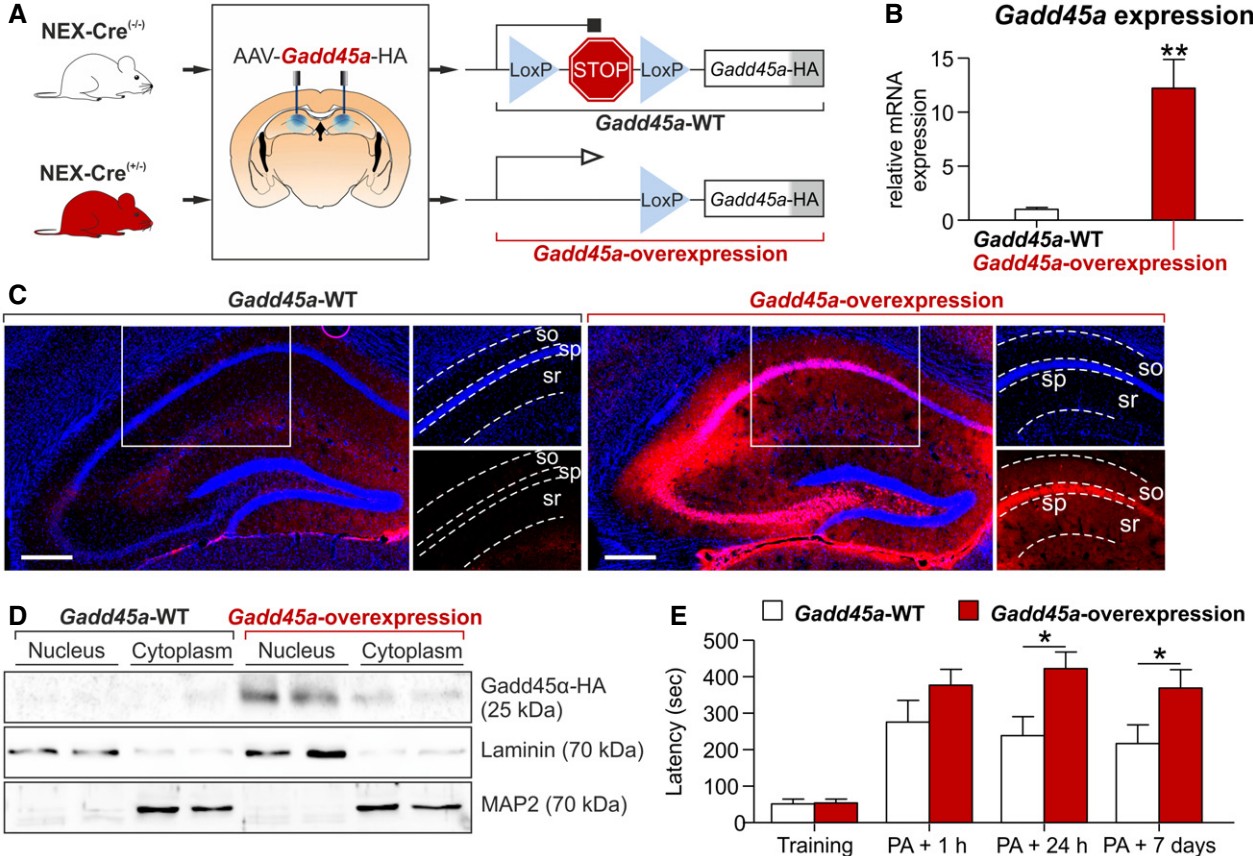

**Figure 2. Gadd45a overexpression enhances memory consolidation.**

A  Diagram of the virally mediated approach to induce *Gadd45a* overexpression in hippocampal glutamatergic neurons of *Nex-Cre*[(+/−)] mice.

B  *Gadd45a* mRNA level in the hippocampus of *Gadd45a*-overexpressing mice (red, n = 7) was significantly increased ($t_{10}$ = 4.189) as compared to *Gadd45a*-WT (n = 5). Values shown are mean ± SEM; unpaired *t*-test: **$P < 0.01$.

C  Immunohistochemistry for hemagglutinin (HA, fused to Gadd45α) in *Gadd45a*-WT (*Nex-Cre*[(−/−)], n = 5) and *Gadd45a*-overexpressing hippocampi (*Nex-Cre*[(+/−)], n = 5) 8 weeks after AAV injection. Note that the Gadd45α-HA signal in *Gadd45a*-overexpressing mice is predominantly found not only in the stratum pyramidale (sp) but also in the stratum oriens and stratum radiatum (so and sr, respectively). Scale bar: 250 μm.

D  Cellular distribution of Gadd45α was analyzed by HA-Western blot on nuclear and cytoplasmic protein fractions of *Gadd45a*-WT- and *Gadd45a*-overexpressing mice (n = 2 for both groups). HA was found predominantly in the nuclear fraction of *Gadd45a*-overexpressing hippocampi, with a smaller but recognizable expression also in the cytoplasm.

E  Latency to enter the dark compartment for *Gadd45a*-WT (white bars, n = 10) and *Gadd45a*-overexpressing mice (red bars, n = 12) in the PA test. A significant increase was found 24 h and 7 days after the PA training in the *Gadd45a*-overexpression group. Values shown are mean ± SEM; two-way ANOVA and Bonferroni post hoc test: *$P < 0.05$.

Source data are available online for this figure.

123 ± 7% (n = 11) at 60 and 90 min, respectively, after LTP induction, which was significantly smaller in both cases ($P < 0.05$, Student's *t*-test) than in *Gadd45a*-WT mice (Fig 3B). In *Gadd45a*-overexpressing animals, the field potential (FP) slope was increased to 168 ± 14% (n = 8) and 166 ± 18% (n = 8) at 60 min and 86 min, respectively. Although not significantly different from that of *Gadd45a*-WT (144 ± 7% and 144 ± 6%, n = 10), the profiles hinted to a tendency to increased synaptic plasticity in *Gadd45a*-overexpressing mice (Fig 3C). The analysis of paired-pulse ratio before and after LTP induction did not show significant differences between any experimental group (Fig EV3A and C), indicating that LTP was mediated by post-synaptic mechanisms. Measurements of synaptic fatigue also provided very similar results in all experimental groups (Fig EV3B and D), therefore excluding that differences in

LTP were caused by insufficient synaptic stimulation. Altogether, these experiments indicate that Gadd45α regulates not only memory consolidation but also its most characteristic functional readout, LTP.

Finally, based on our observation of deficient memory consolidation of *Gadd45a*-KO mice in cued fear conditioning, a process that relies on an intact amygdala [22], we performed LTP studies also in this brain area (Fig EV3E), observing a strongly decreased LTP in the lateral amygdala of *Gadd45a*-KO as compared to *Gadd45a*-WT (Fig EV3F), while the paired-pulse ratio of the synaptic FP responses and synaptic fatigue was not altered (Fig EV3G and H). These results demonstrate that the effects of Gadd45α on learning and synaptic plasticity are not only restricted to the hippocampus, but also include the amygdala.

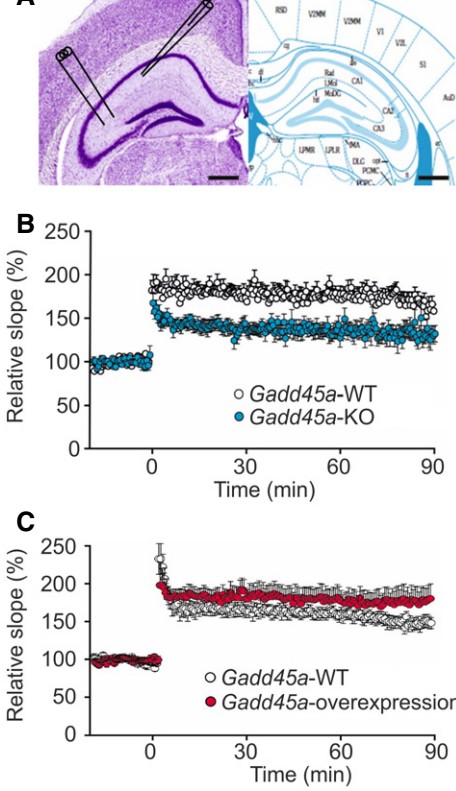

**Figure 3.  *Gadd45a* regulates hippocampal synaptic plasticity.**

A  Nissl-stained slice (left) and corresponding section from brain atlas (right). Left panel shows hippocampus and position of the stimulating (left) and recording (right) electrode. Scale bars: 500 μm (black bar).

B  LTP protocol was applied to 16 slices of *Gadd45a*-WT mice (white dots, *n* = 5 animals) and 11 slices of *Gadd45a*-KO mice (blue dots, *n* = 5 animals), resulting in a significantly (*P* < 0.05, Student's *t*-test) lowered induction of LTP in *Gadd45a*-KO mice as compared with *Gadd45a*-WT. Values shown are mean ± SEM.

C  *Gadd45a*-overexpressing mice (red dots, *n* = 8 slices from 6 animals) presented a slightly increased LTP as compared with *Gadd45a*-WT (white dots, *n* = 10 slices from 6 animals), which became more pronounced during the late phase of the experiment. Values shown are mean ± SEM.

## Gadd45α affects the expression of a distinct set of memory-related genes

To elucidate the molecular mechanism underlying the deficiency in memory consolidation and synaptic plasticity in *Gadd45a*-KO mice, we carried out a genome-wide transcriptome analysis by poly(A) RNA-seq, using hippocampal samples of *Gadd45a*-WT (*n* = 6) and *Gadd45a*-KO mice (*n* = 6) collected 1 h after the training in the PA paradigm. Principal component analysis shows sample clustering based on the genotype (Appendix Fig S1A). Differential expression analysis revealed approximately 400 deregulated genes, almost all being down-regulated in the *Gadd45a*-KO samples as compared to wild-type (Fig 4A, Dataset EV1). Gene ontology analysis of the deregulated genes revealed an enrichment in genes important for memory and synaptic function (Fig 4B, Dataset EV2). Interestingly, 73 of the top 100 down-regulated genes showed an extended 3′UTR

(continuous read coverage for at least 2 kb beyond the canonical polyadenylation site according to the Ensembl gene annotation of the mouse genome NCBIM37/mm9) (Fig 4C). Among these genes, *Grin2a* (coding for the subunit epsilon 1 of the ionotropic glutamate NMDA receptor) is of particular interest due to the critical role of this receptor type in memory and synaptic plasticity [23–26]. Genome browser inspection showed not only a decreased *Grin2a* exonic coverage in *Gadd45a*-KO mice (Fig 4D), but also an unusual lengthening of the 3′UTR, which apparently extends the transcriptional termination far beyond the canonical polyadenylation site (Fig 4E). Such extended 3′UTRs have recently been reported particularly in neural tissues, including hippocampus [27]. Importantly, the length of these long 3′UTRs does not differ between genotypes, indicating that Gadd45α is not involved in the 3′UTR elongation of this set of transcripts.

Analysis of this very long non-canonical 3′UTR of *Grin2a* mRNA showed a clear 5′<3′ gradients of RNA-seq read coverage, which was highly reproducible between replicates (Appendix Fig S2), but differed in their steepness between the *Gadd45a*-WT and *Gadd45a*-KO sample groups (Fig 4E). As a result, the read coverage differences in *Grin2a* exons between *Gadd45a*-WT and *Gadd45a*-KO are clearly observable throughout the protein-coding sequence, and at the beginning of the 3′UTR but not at the 3′ end of the extended (non-canonical) 3′UTR. Taking into account the poly(A) selection procedure used in RNA-seq, the fact that read coverage at the end of the 3′UTR is very similar in both groups suggests that transcription is not affected in the absence of Gadd45α. This observation implicates the putative existence of a Gadd45α-dependent post-transcriptional mechanism that regulates mRNA levels rather by promoting mRNA stability than affecting gene transcription. Similar alterations were found for other Gadd45α-regulated targets, such as *Grin2b* (ionotropic glutamate receptor, subunit epsilon 2), *Kcnq3* (potassium voltage-gated channel, subfamily Q, member 3), and *Grm5* (metabotropic glutamate receptor 5) (Appendix Fig S3A–F). Of note, these genes have been previously implicated in memory formation and synaptic plasticity [28–34].

The 5′<3′ read coverage gradients *per se* can be partially of technical nature. Since the employed TruSeq mRNA-sequencing involved poly(A) selection, read coverage is expected to decrease with increasing distance from the transcript 3′ end upon partial RNA fragmentation during sample preparation. However, the consistently reduced steepness of this coverage gradient in KO as compared to WT is indicative of increased transcript instability associated with Gadd45α deficiency. A purely technical artifact explanation is unlikely since (i) RNAs were prepared and sequenced in a randomized fashion, making batch effects unlikely; (ii) overall RNA integrity, concentration, and purity were similar among all samples; (iii) increased transcript fragmentation as inferred from the RNA-seq coverage tracks was observed in all six samples of the *Gadd45a*-KO group (Appendix Fig S2); (iv) overall 5′–3′ read coverage profiles are similar between genotypes (Appendix Fig S1B); and (v) a number of genes with extended 3′UTRs, but without annotated functions regarding memory and synaptic plasticity, did not show the 5′<3′ gradient and are not differentially expressed between *Gadd45a*-WT and *Gadd45a*-KO (e.g., *Map2k6*; Fig EV4A and B).

To validate these findings, we performed qPCR for *Grin2a* (TaqMan assay covering exons 3–4) on RNA extracted from hippocampus after 1 h following PA training, using cDNA

reverse-transcribed either with random primers or with oligo(dT) primers (priming the poly(A) tail at the 3′ end of mRNAs). These two alternative strategies render two distinct pools of cDNA. While all mRNAs are reverse-transcribed when using random primers (independently of their stability/fragmentation), only polyadenylated mRNAs are reverse-transcribed when using oligo(dT) primers

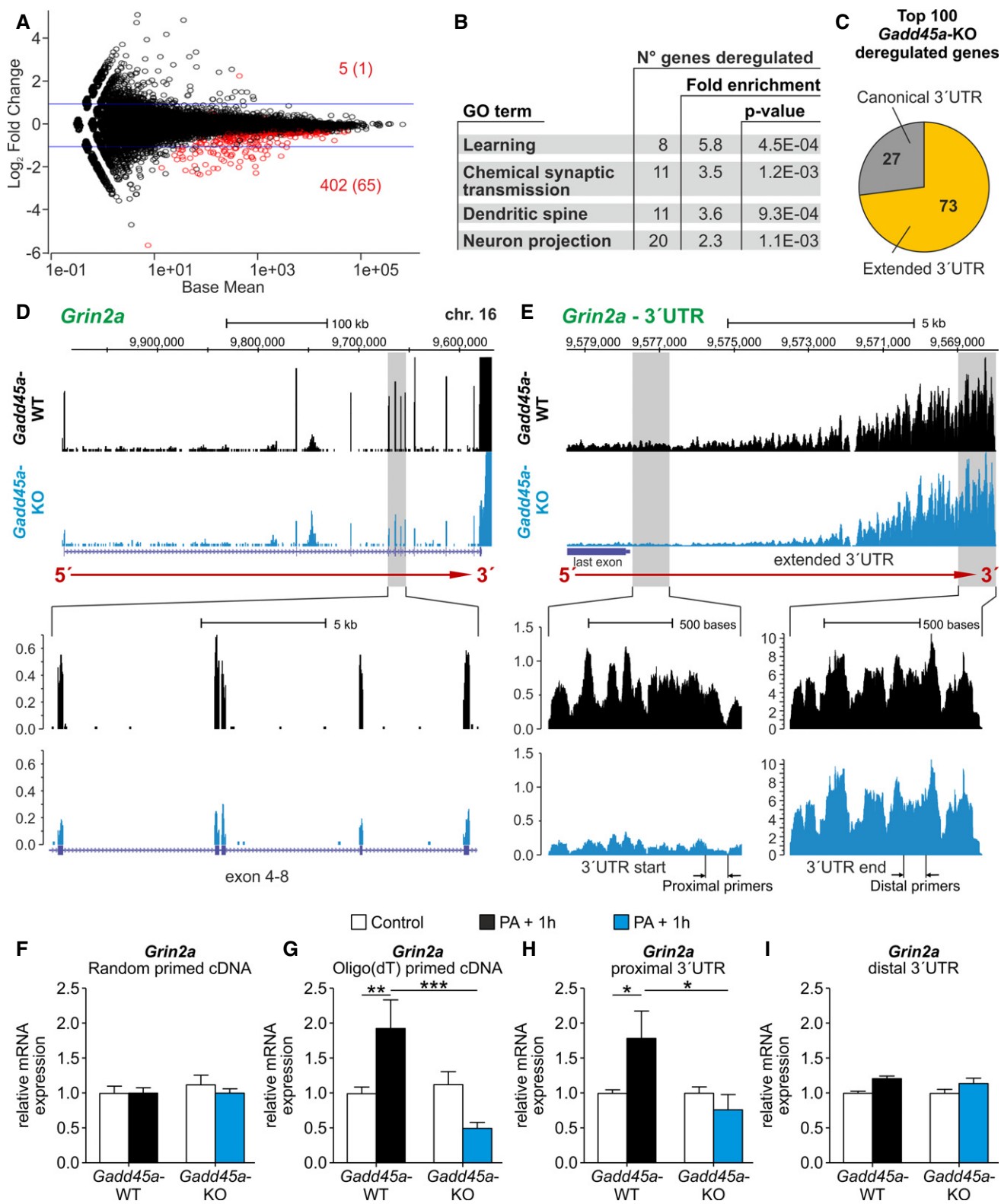

Figure 4.

**Figure 4.  Gadd45a affects the expression of a particular set of memory-related genes.** RNA-seq was performed on *Gadd45a*-WT (*n* = 6) and *Gadd45a*-KO hippocampi (*n* = 6). Samples were collected 1 h after exposure to PA.

A    MA plot of DESeq differential expression analysis with the 1% false discovery rate (FDR) hits in red (5/402 and with > 2-fold change 1/65 up-/down-regulated, respectively). The affected genes are described in Dataset EV1.

B    Gene ontology (GO) enrichment analysis of the 402 down-regulated genes revealed gene classes related to memory formation and synaptic function. For a complete analysis of gene ontology, see Dataset EV2.

C    Proportion of transcripts with extended 3′UTRs (at least 2 kb beyond the canonical polyadenylation site) among the top 100 Gadd45α hits.

D    UCSC browser view of reads-per-million-normalized RNA-seq coverage for *Grin2a* in *Gadd45a*-WT (black) and *Gadd45a*-KO (blue) representative samples. Gene structure of *Grin2a* (colored in purple) and the direction of transcription (red arrow) are depicted. Upper panel covers the entire transcript while lower panel is an enlargement of exons 4–8.

E    Detailed view of the extended 3′UTR of *Grin2a* showing a significant 5′<3′ gradient in RNA-seq coverage. Note that the normalized read coverage is similar between genotypes at the actual 3′UTR end but differs significantly upstream including the annotated exons. For subsequent experiments, position of the designed primers covering proximal and distal parts of the 3′UTR is indicated below.

F, G    Validation of Gadd45α-regulated levels of *Grin2a* mRNA by qPCR with primers covering exons 3–4. *Grin2a* mRNA levels were analyzed in the hippocampus of mice under control conditions (white bars: *Gadd45a*-WT = 7 and *Gadd45a*-KO = 7) and 1 h after PA (*Gadd45a*-WT = 5, black bars; *Gadd45a*-KO = 6, blue bars). Random-primed cDNA levels (F) were very similar in all the experimental groups. Oligo(dT)-primed cDNA (G) revealed a significant increase in *Grin2a* 1 h after PA, which was only observed in *Gadd45a*-WT mice. Values shown are mean ± SEM; two-way ANOVA and Bonferroni post hoc test: **$P < 0.01$, ***$P < 0.001$.

H, I    mRNA levels of proximal versus distal parts of the 3′UTR analyzed in the same set of oligo(dT)-primed cDNA. (H) Significant increase in *Grin2a* mRNA levels was found for the proximal part only in *Gadd45a*-WT. (I) Distal parts of the 3′UTR remained unchanged (*n* = 6 for all groups). Values shown are mean ± SEM; two-way ANOVA and Bonferroni post hoc test: *$P < 0.05$.

(excluding all fragmented and hence non-polyadenylated mRNAs) (Appendix Fig S4). Consistent with the mRNA stability hypothesis (no transcriptional change, but post-transcriptional stabilization), no significant differences in *Grin2a* levels were observed in the randomly reverse-transcribed samples (Fig 4F). This implies that the total level of *Grin2a* transcription remained unaltered between the different experimental groups. In contrast, qPCR with oligo(dT) reverse-transcribed cDNA revealed an induction of *Grin2a* 1 h after the initial training in the PA test ($t_{13} = 3.387$, $P < 0.01$; Fig 4G), which was impaired in *Gadd45a*-deficient mice ($t_{12} = 4.850$, $P < 0.001$). Moreover, similar results were obtained for other Gadd45α targets identified above, such as *Grin2b*, *Kcnq3,* and *Grm5* (Appendix Fig S5A and B). To corroborate these findings and exclude potential batch-dependent effects, we analyzed an independent set of *Gadd45a*-WT and *Gadd45a*-KO samples obtaining similar expression profiles (Appendix Fig S5C and D).

Of note, analysis of exonic and intronic read coverage was recently used to differentiate between transcriptional and post-transcriptional regulation [35]. Preferential changes in the read coverage of exons over introns can be attributed to post-transcriptional rather than transcriptional regulation [35]. This is the case for *Grin2a*, *Grm5, Grin2b,* and *Kcnq3* (Fig EV4C). Furthermore, differential expression analysis at the exon level revealed that most exons contained in the affected transcripts are significantly down-regulated in *Gadd45a*-KO samples (Appendix Fig S6A, Appendix Table S1). Moreover, pre-mRNA analysis by qPCR using primers covering intronic regions showed no differences between the experimental groups (Fig EV4D) for *Grin2a* and *Grm5*, further corroborating that transcription is not affected in the absence of Gadd45α.

Taken together, the above results suggest a transcription-independent mRNA regulation by Gadd45α that involves changes in mRNA stability. Thus, we aimed at corroborating the differential stability of the *Grin2a* transcript (5′ versus 3′) by performing qPCR on distinct segments of the extended 3′UTR. We reverse-transcribed RNA only with oligo(dT) primers and amplified different *Grin2a* regions. By using primers covering proximal or distal parts of the 3′UTR (Fig 4E), we observed that on the one hand the distal part of the *Grin2a* 3′UTR was not significantly different between genotypes and experimental groups, indicating the absence of differences at

the transcriptional rate (Fig 4I). On the other hand, proximal parts of the extended 3′UTR were enhanced only in *Gadd45a*-WT hippocampi after PA training ($t_{10} = 2.688$, $P < 0.05$; Fig 4H). Similar results were found for *Grm5* (Appendix Fig S6B) and also for *Grin2a* and *Grm5* in the independent set of samples previously mentioned (Appendix Fig S6C and D). Considering the absence of differences at the transcriptional level, these findings point to a Gadd45α-dependent mechanism of post-transcriptional RNA stabilization during memory consolidation.

Finally, given the opposite phenotype that was observed in *Gadd45a*-overexpressing mice (i.e., increased memory and synaptic plasticity), we also evaluated the expression of *Grin2a* and *Grm5* in the hippocampi of these mice. qPCR analyses revealed enhanced levels of the two transcripts, both under controls conditions and 1 h after PA, as compared to *Gadd45a*-WT mice (Fig EV4E and F). Interestingly, in the case of *Grin2a*, this difference became statistically significant only when analyzing poly(A) mRNAs (oligo(dT) primed). Therefore, *Gadd45a* overexpression not only improved memory consolidation and LTP, but also enhanced the expression of memory-related genes.

## Gadd45α controls synaptosomal NR2A protein levels and interacts with NR2A encoding mRNA

In order to evaluate the consequences of the observed destabilization of *Grin2a* mRNAs in *Gadd45a*-KO at the protein level, we quantified NR2A protein levels (encoded by *Grin2a*) in total and synaptosomal protein fractions of the hippocampus. By using a protocol designed to specifically isolate membrane synaptosomal proteins, such as NR2A and post-synaptic density protein 95 (PSD95) (Fig EV5A), we detected activity-dependent NR2A membrane insertion (Fig 5A). Of note, this process has been previously shown to depend on local translation to NR2A, which is in turn controlled by the 3′UTR of the *Grin2a* transcript [3]. Importantly, total levels of NR2A remained unchanged during memory formation, but synaptosomal NR2A was significantly increased only in *Gadd45a*-WT hippocampi ($t_{23} = 3.121$, $P < 0.01$; Fig 5A and B). This was also observed for mGluR5 (encoded by *Grm5*) ($t_{22} = 3.199$, $P < 0.01$; Fig EV5B and C). Thus, the

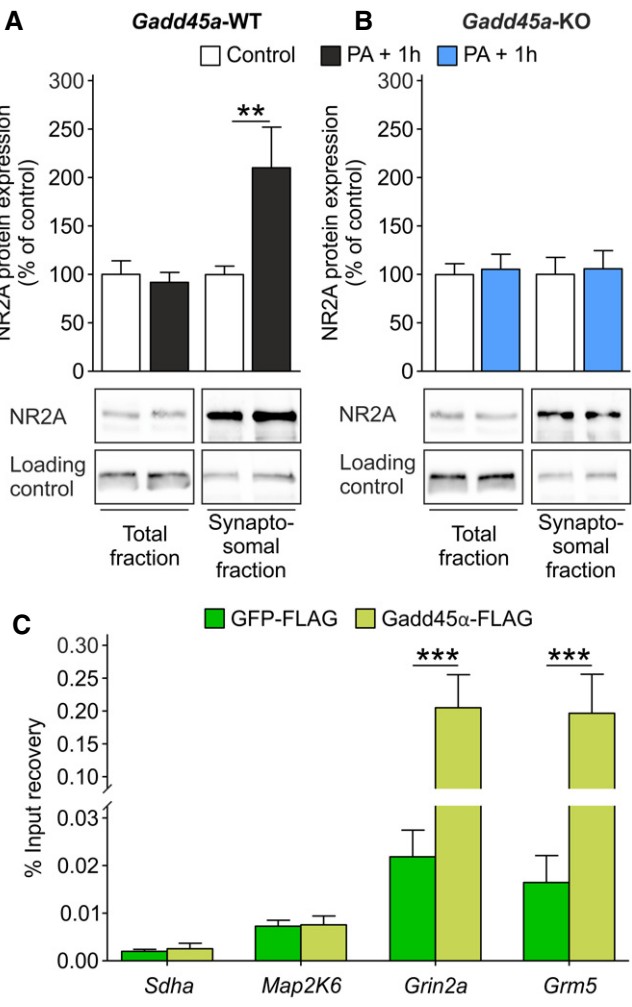

**Figure 5. Gadd45α regulates synaptosomal NMDA receptor (NR2A) levels and specifically interacts with NR2A mRNA (*Grin2a*).**

A, B  NR2A protein levels were analyzed in the hippocampus of mice under control conditions (white bars: *Gadd45a*-WT, n = 13; *Gadd45a*-KO, n = 14) and 1 h after PA (black bars: *Gadd45a*-WT = 14, *Gadd45a*-KO = 14). NR2A signal (160 kDa) was normalized to the loading control (tubulin (55 kDa) for the total fraction and synaptotagmin (68 kDa) for the synaptosomal fraction) and calculated as percentage of control. In *Gadd45a*-WT samples (A), NR2A protein levels were unchanged in the total fraction during memory formation, while in the synaptosomal fraction, a significant increase was detected ($t_{20}$ = 3.199). In *Gadd45a*-KO hippocampi (B), levels of NR2A protein in both fractions remained unaltered during memory consolidation. Values shown are mean ± SEM; two-way ANOVA and Bonferroni post hoc test: **$P < 0.01$.

C  Gadd45α pulldown experiments were carried out in wild-type hippocampal lysates (n = 12 for all groups). Percentage of input recovery was unchanged between GFP-FLAG (green bars) and Gadd45α-FLAG (black bars) for genes that are not regulated by Gadd45α, such as succinate dehydrogenase complex flavoprotein subunit A (*Sdha*, no 3′UTR extension), and mitogen-activated protein kinase 6 (*Map2K6*, containing 3′UTR extension of 10 kb). On the contrary, Gadd45α-FLAG-bound fraction revealed a significant increase in *Grin2a* ($t_{22}$ = 4.992) and *Grm5* mRNA ($t_{22}$ = 4.939) as compared to GFP-FLAG fraction. Values shown are mean ± SEM; two-way ANOVA and Bonferroni post hoc test: ***$P < 0.001$.

Source data are available online for this figure.

post-transcriptional stabilization of the *Grin2a* and *Grm5* transcripts by Gadd45α promotes the increase in NR2A and mGluR5 proteins in synaptosomal membranes. Additionally, we also measured the levels of *Grin2a* and *Grm5* mRNAs in the synaptosomal fractions, obtaining a significant reduction in *Gadd45*-KO samples in both control and PA + 1 h groups (*Grin2a*, $F_{1,20}$ = 13.55, $P < 0.01$; *Grm5*, $F_{1,20}$ = 23.9, $P < 0.001$) (Fig EV5D). These results suggest that Gadd45α modulates the expression of these proteins by affecting the localization of their respective mRNAs.

Next, we aimed at elucidating the underlying mechanism by which Gadd45α regulates the stability of *Grin2a* and *Grm5* transcripts. Since Gadd45α is an RNA-binding protein [9], we tested the specificity of this binding to the Gadd45α targets identified above in comparison with transcripts which were not regulated by Gadd45α (e.g., *Sdha*, succinate dehydrogenase complex flavoprotein subunit A; *Map2k6*, mitogen-activated protein kinase 6). Incubating immobilized Gadd45α and GFP (a protein without RNA-binding capacity as control), respectively, with mouse hippocampal lysates, we observed a significant enrichment of *Grin2a* ($t_{22}$ = 4.992, $P < 0.001$) and *Grm5* ($t_{22}$ = 4.939, $P < 0.001$) mRNA in the Gadd45α-bound fraction (Fig 5C). This was also true for other targets of Gadd45α, such as *Grin2b* ($t_{22}$ = 6.861, $P < 0.001$) and *Kcnq3* ($t_{22}$ = 3.267, $P < 0.01$) (Fig EV5G). Importantly, Map2k6 was not enriched in the Gadd45α-bound fraction despite having a 3′UTR extension comparable with that present in Gadd45α target mRNAs (Fig EV4A), strongly indicating the target specificity of Gadd45α, and suggesting a mechanism underlying specific mRNA stabilization by the Gadd45α protein in the process of memory consolidation.

## The RNA-binding capacity of Gadd45α is required for the increased memory in a model of Gadd45α overexpression

Finally, we aimed at elucidating the importance of the RNA-binding capacity of Gadd45α for the memory improvement observed in *Gadd45a*-overexpressing mice. It was recently demonstrated that lysine 45 is essential for the protein to bind RNA [9,36]. The substitution of lysine for the negatively charged residue glutamate impedes the capacity of Gadd45α to bind the negatively charged RNA. Therefore, we generated an AAV containing the K45E mutant form of Gadd45α, which was then injected into the dorsal hippocampus of *Nex-Cre*[(+/−)] mice (Fig 6A). This yielded a 15-fold overexpression at the mRNA level (Fig 6B) and also to overexpression at the protein level (Fig 6C). HA immunostaining on hippocampal sections from these mice revealed a similar protein distribution as compared to the wild-type form of the AAV-*Gadd45a* (Fig 6D). Higher magnification pictures showed that the Gadd45α-HA signal was not restricted to the nucleus but was also present in neuronal bodies and projections (Fig 6D), confirming the cellular distribution of Gadd45α described above (Fig 2D). Interestingly, when *Gadd45a*[K45E]-overexpression mice were exposed to the PA, they failed to improve memory retention as compared to *Gadd45a*-overexpression mice (Fig 6E). This was true for both 1 h ($t_{31}$ = 3.747, $P < 0.05$) and 24 h after the PA training ($t_{31}$ = 3.817, $P < 0.05$). In fact, their latency to explore the dark compartment was very similar to that of *Gadd45a*-WT mice during the entire experiment (Fig 6E). These results indicate that Gadd45α needs to bind RNA in order to improve learning in a model of *Gadd45a* overexpression, demonstrating the importance of this protein in mRNA regulation.

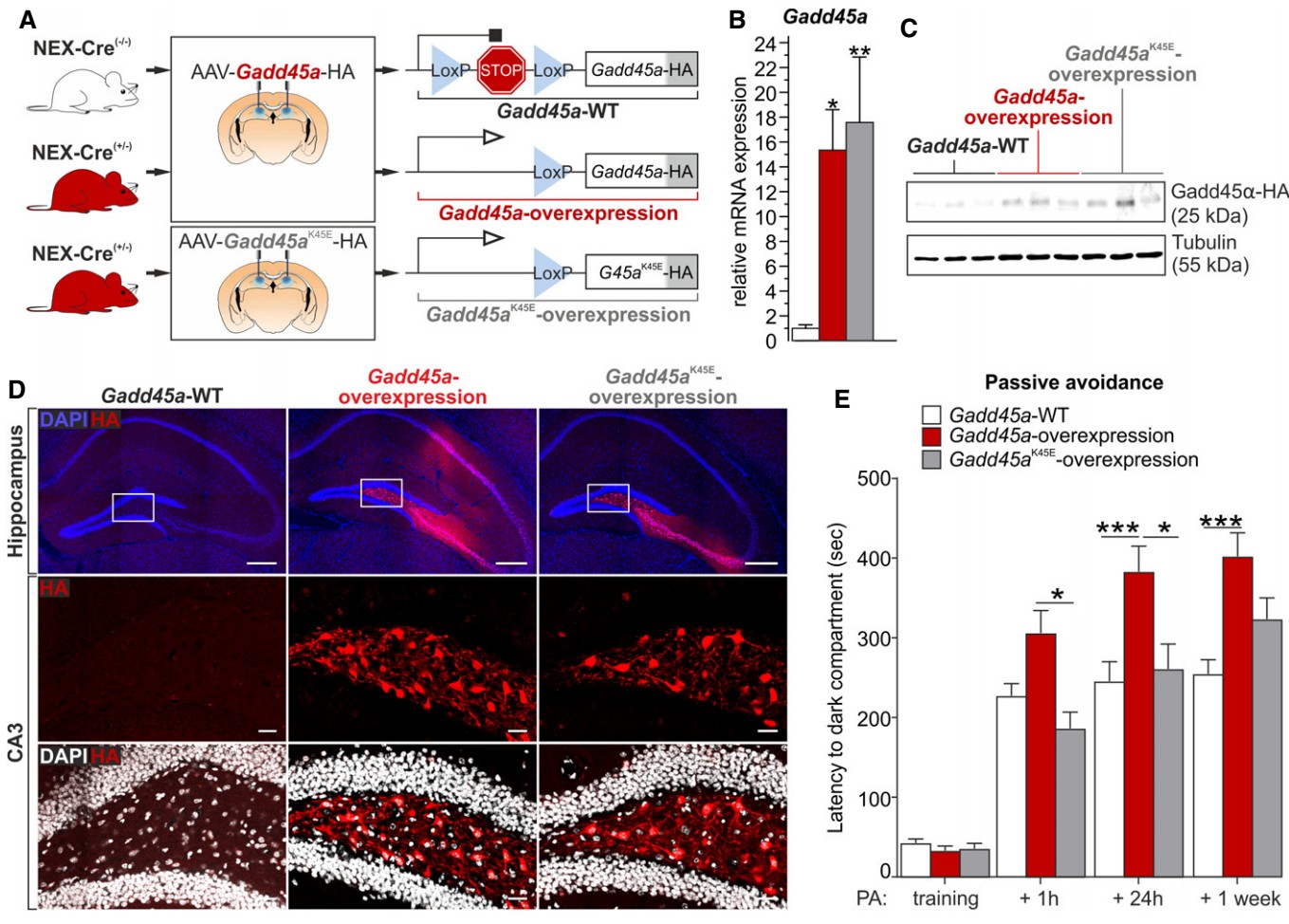

**Figure 6. Gadd45α requires its RNA-binding capacity to mediate its action on memory.**

A Diagram of the virally mediated approach to induce *Gadd45a*-overexpression in hippocampal glutamatergic neurons of *Nex-Cre*[(+/−)] mice. On the one hand, *Nex-Cre*[(−/−)] (n = 12) and *Nex-Cre*[(+/−)] (n = 11) were injected with the AAV-*Gadd45a*-HA used in Fig 2. On the other hand, a group of *Nex-Cre*[(+/−)] (n = 11) was injected with a mutated form of Gadd45α (K45E) that lacks the RNA-binding capacity.

B Hippocampal *Gadd45a* mRNA levels were significantly up-regulated in both *Gadd45a*-overexpression ($q_{24}$ = 4.094) and *Gadd45a*[K45E]-overexpression groups ($q_{24}$ = 4.726) (n = 9 for all groups). Values shown are mean ± SEM; Tukey's multiple comparison test: *$P < 0.05$, **$P < 0.01$.

C Gadd45α overexpression was confirmed at the protein level by HA-Western blot on hippocampal samples from *Gadd45a*-WT, *Gadd45a*-overexpression, and *Gadd45a*[K45E]-overexpression mice (n = 3 for all groups).

D Immunohistochemistry for hemagglutinin (HA, fused to Gadd45α) in *Gadd45a*-WT (left), *Gadd45a*-overexpression (middle) and *Gadd45a*[K45E]-overexpression (right) samples 8 weeks after AAV injection (n = 3 for all groups). Upper row depicts the entire hippocampus, while center and lower rows show magnifications of the CA3 area in which the HA signal (red) is not restricted to the nuclei (blue and white). Hippocampus scale bars: 250 μm. Hilus scale bars: 50 μm.

E Latency to enter the dark compartment for *Gadd45a*-WT (white bars, n = 12), *Gadd45a*-overexpression (red bars, n = 11), and *Gadd45a*[K45E]-overexpression mice (gray bars, n = 11) in the PA test. Note that, unlike *Gadd45a*-overexpression mice, those expressing the mutated form of Gadd45α (no RNA-binding capacity) did not show an increase in memory retention. Values shown are mean ± SEM; two-way ANOVA and Bonferroni post hoc test: *$P < 0.05$, ***$P < 0.001$.

Source data are available online for this figure.

## Discussion

Our study demonstrates for the first time a neural function for Gadd45α specifically in the consolidation of aversive memory and LTP, thereby contributing to the understanding of memory processing. Additionally, it also offers a new perspective on a post-transcriptional mechanism involving extended 3′UTRs. Specifically, Gadd45α contributes to the maintenance of RNA stability of specific memory-related genes during memory consolidation. In the absence of Gadd45α, these transcripts, which contain extended 3′UTRs, become destabilized, resulting in reduced synaptosomal mRNA levels and expression of the corresponding encoded proteins, and leading to deficits in synaptic plasticity and memory consolidation.

Despite strong expression of Gadd45α during different stages of development [10,37], most of the basic behaviors analyzed were not significantly compromised by life-long *Gadd45a* deficiency (Fig EV1). The functions of the Gadd45 family members are similar, but not identical, and their induction differs depending on physiological conditions and cell types [38]. Moreover, Gadd45 proteins can interact with each other, and these interactions may play a

pivotal role in their functions [38]. In fact, studies on the role of Gadd45 proteins during neural development suggest compensatory processes [39]. In this context, it is not surprising that *Gadd45a*-KO mice display unaltered behavior in most of the tasks analyzed and that Sultan *et al* observed mostly normal baseline behavioral tasks in *Gadd45b*-KO mice [12]. However, a distinct set of behaviors characterized by aversive learning was impaired in *Gadd45a*-KO mice. We found that *Gadd45a*-KO mice performed worse in context-dependent associative learning (PA; Fig 1B) and in spatial navigation (WCM; Fig EV1F), two-well characterized behaviors where the hippocampus plays a pivotal role [40,41]. Together with the increased learning observed in *Gadd45a*-overexpressing mice, our results demonstrated a novel role for Gadd45α specifically in the regulation of memory consolidation. This relates well with previous findings regarding the role of Gadd45β [11,12] and Gadd45γ [42] in memory formation.

In line with the behavioral phenotypes, LTP was decreased in *Gadd45a*-KO mice and enhanced in *Gadd45a*-overexpressing animals (Fig 3B and C). Glutamate receptors are essential for this type of synaptic plasticity [26,43]. In *Gadd45a*-KO mice, the levels of *Grin2a* and *Grm5* mRNA in synaptosomes were significantly reduced under basal conditions (without PA exposure) (Fig EV5D). This suggests a basal state in which the capacity to form LTP is compromised before the induction of aversive learning. Moreover, the same two transcripts were also significantly elevated in *Gadd45a*-overexpressing mice prior to PA exposure, thus enhancing the capacity to form LTP (Fig EV4E and F). Additionally, ion channels are important players in the formation of LTP [44], and, congruently, in *Gadd45a*-KO mice, there was a reduction in the expression of six subunits of potassium voltage-gated channels (*Kcna2*, *Kcna3*, *Kcnma1*, *Kcnk9*, *Kcng3*, and *Kcnq3*), one subunit of calcium voltage-gated channels (*Cacna1e*), and two subunits of transient receptor cation channels (*Trpc5 and Trpm3*) (Dataset EV1). Thus, these alterations in the ion channel repertoire in neurons can also impair neuronal information processing.

We discovered a remarkable feature in the transcriptome of *Gadd45a*-KO mice where the great majority of the down-regulated mRNAs at 1 h after PA training contained long extensions of the 3′UTR, which are characteristic for neurons and are particularly abundant in hippocampus as compared to other brain regions and non-neuronal tissues [27,45]. Of note, the length of these extensions is not Gadd45α-dependent. Sequence elements located in the 3′UTR are targets for multiple post-transcriptional control mechanisms, including microRNA (miRNA)-mediated translational suppression, RNA stabilization/destabilization, subcellular RNA localization, and translational efficiency. 3′UTR extensions can provide a remarkable landscape for miRNA-dependent regulation. miRNAs can function post-transcriptionally by base-pairing to the 3′UTR to repress translation by mechanisms that are not fully understood yet [46]. miRNAs are known to modulate a number of brain functions, such as neuronal maturation, memory formation, and synaptic plasticity [47]. In fact, the turnover of miRNAs is dynamically regulated by neuronal activation [48], and, e.g., Gadd45α has been shown to activate miRNA-295 during somatic cell reprogramming [49]. Of note, the discovered impairments in the expression of ion channels in *Gadd45a*-KO mice might be caused by miRNA action on the 3′UTR extensions of these mRNAs [50]. Importantly, regulation of miRNA-mediated repression by

RBPs is a widespread mechanism [46]. Therefore, Gadd45α could control the functionality of memory-related transcripts by regulating the availability and activity of miRNAs. Regarding RNA stabilization/destabilization, RNAs with long 3′UTRs are frequently targeted by the nonsense-mediated mRNA decay pathway (NMD) [51,52], which is a crucial post-transcriptional regulatory event that is functionally incorporated into homeostatic mechanisms in response to environmental cues [53]. This highly conserved mechanism for surveillance of mRNA stability is characterized by endonucleolytic cleavage of processed mRNAs with premature termination codons [54]. In this context, UP-Frameshift factors (UPF1, UPF2, and UPF3) are key players in the induction of NMD [55]. In fact, UPF1 associates with mRNAs in a 3′UTR length-dependent manner to potentiate mRNA decay [56]. Of note, UPF3-deficient mice were reported to show a reduced memory in cued and contextual fear learning tasks [57], demonstrating the importance of NMD in learning and memory. Interestingly, it is believed that after endonucleolytic cleavage by NMD, exonuclease-mediated degradation of the primary cleavage products can be temporally inhibited, allowing the detection of fragmented mRNAs [58]. Congruently, in our experimental design, detection of all the fragmented mRNAs independently of the presence of a poly(A) tail (using random-primed cDNA) prevented the detection of genotype differences. On the contrary, detection of only polyadenylated mRNAs (using oligo(dT) primed cDNA) ensured the proper discrimination between transcriptional and post-transcriptional (i.e., NMD-mediated) effects (Appendix Fig S4).

Given the absence of differences in total mRNA (Fig 4F), as well as in pre-mRNA levels (Fig EV4D) and in read coverage of introns (Fig EV4C), we can conclude that Gadd45α does not regulate the transcriptional rate of its targets. Considering the role of Gadd45α in active DNA demethylation, the lack of effects at the transcriptional level is somewhat surprising. However, Gadd45 proteins might act redundantly in DNA demethylation *in vivo*, and loss of one isoform might be compensated by the other cognate genes [59]. Indeed, in zebrafish, only combined knockdown of all Gadd45 isoforms impairs DNA demethylation [60]. Read coverage inspection of *Bdnf*, a well-known modulator of hippocampal memory and target of DNA methylation/demethylation processes, did not show any significant difference between genotypes in our RNA-seq data. Moreover, the majority of Gadd45α targets identified in the RNA-seq experiment revealed a similar read coverage in both genotypes toward the end of the transcripts (3′ end of the 3′UTR). As discussed above, this feature indicates the absence of Gadd45α-dependent regulation at the transcriptional level.

This prompted us to explore alternative explanations. Alternative cleavage and polyadenylation (APA) is a process taken into consideration when studying extensive lengthening of mRNAs. The usage of different polyadenylation sites within the 3′UTR is relatively common, producing a variety of transcripts that differ in their length (short versus long transcripts; proximal versus distal polyadenylation sites, respectively), and eventually in their functions. For instance, *Bdnf* expresses an extended 3′UTR isoform, which localizes to dendrites, and a short 3′UTR APA variant that is restricted to the soma [61]. However, when studying APA events via RNA-seq, two important characteristics should be mentioned. First, APA typically creates step-wise changes in read coverage at the APA site. Second, these changes result in lower read coverage toward the 3′

end of the transcript. Importantly, none of these two basic features of APA are seen in Gadd45α-targets. Thus, APA cannot explain the Gadd45α-dependent profile of gene expression observed after memory formation. Nevertheless, formal exploration of the APA hypothesis using the published tool DaPars [62] revealed no significant Gadd45α-dependent APA events.

Our findings demonstrate a new Gadd45α-dependent mechanism whereby the availability of functional mRNA is regulated by a post-transcriptional mechanism. Our analyses of distal versus proximal parts of the *Grin2a and Grm5* 3′UTRs revealed a process in which only the proximal parts of these transcripts were increased upon memory consolidation (Fig 4H and I). Considering that the total amount of the transcripts is the same, the most plausible explanation is an increase in stability of the pre-existing transcripts and, consequently, an increase in the corresponding mRNA levels. Similar processes have already been described for *Bdnf* [2,63].

Most of the post-transcriptional mechanisms mentioned above are directly or indirectly regulated by different RBPs [45]. Recently, Gadd45α has also been described as a RBP, with similar structural–functional features as other members of the L7Ae/L30e/S12e RBP superfamily [9]. Among these features, the binding to the kink-turn motif (a recurrent architectural RNA element) and the presence of Gadd45α in nuclear speckles are of special interest in understanding the proposed post-transcriptional mechanism of Gadd45α. In fact, kink-turn motifs have been suggested as mediators of the long-distance transport of RNAs to different subcellular locations [64]. This process is present in neurons to ensure local translation, which is essential for synaptic plasticity and memory formation. For example, it was shown that NMDA receptors are locally translated during memory formation [65]. Along this line, we observed increased levels of NR2A and mGluR5 proteins during memory consolidation in the synaptosomes of *Gadd45a*-WT, but not in those from *Gadd45a*-KO. This is consistent with our observation of impaired stabilization of *Grin2a* mRNA in the absence of Gadd45α and consequent memory deficit. Interestingly, *Grin2a* and *Grm5* mRNAs were significantly reduced in the synaptosomes from *Gadd45a*-KO, suggesting that Gadd45α controls the trafficking of memory-related transcripts. Considering the remarkable compartmentalization of the neuron, a change in mRNA localization could have an important impact on the extent of miRNA-dependent repression and/or local translation of these transcripts. In agreement with such a model, activity-induced local translation of *Grin2a* was shown to depend on its 3′UTR [3]. In fact, long 3′UTR forms of the mRNA for cluster of differentiation 47 protein (CD47) have been described as regulators of activity-induced membrane localization [66]. In this study, the scaffold function of the long 3′UTR (as opposed to the short form) directs the transport and function of CD47 during translation [66]. Alternatively, Gadd45α could also affect the stability of memory-related genes in specific neuronal compartments, by controlling the cellular distribution and/or local activity of the miRNA repression machinery.

Furthermore, the presence of Gadd45α in nuclear speckles supports the notion of Gadd45α as a mediator of post-transcriptional regulatory mechanisms, since nuclear speckles are also sites of RNA splicing and processing [9]. We also showed that the Gadd45α-HA protein detected by HA immunostainings in the *Gadd45a*-overexpression model is not restricted to the nucleus. Additionally, by performing nuclear/cytosolic fractionation in the same samples, we could detect Gadd45α-HA in the cytoplasm, where most of the post-transcriptional regulation occurs. This leaves open the question regarding the specific cellular compartment in which Gadd45α mediates its effects on mRNA stability. In this context, another center for active mRNA modifications is RNA granules, which are large accumulations of mRNAs and RBPs. Interestingly, Han *et al* [67] demonstrated that RNAs with extended 3′UTRs (including *Grin2a*) are present in RNA granules.

Finally, our validation of the RNA-binding capacity of Gadd45α specifically to memory-related transcripts containing extended 3′UTRs offers a direct link between Gadd45α function and the post-transcriptional mechanisms required for memory formation and synaptic plasticity. Additionally, the use of a mutated form of *Gadd45a* lacking the RNA-binding capacity (*Gadd45a*[K45E]-overexpressing mice) allowed us to corroborate *in vivo* the importance of the association between the Gadd45α protein and its mRNA targets.

In summary, this study sheds light on the poorly understood biological role of extended 3′UTRs by implicating them for the first time in aversive memory consolidation and LTP. Importantly, the identification of Gadd45α as a critical regulator of 3′UTR stability in memory formation provides a molecular handle with which this process can now be further unraveled.

# Materials and Methods

### Animals

Adult male mice (2–6 months) were socially housed in a temperature- and humidity-controlled room with a 12- to 12-h light–dark cycle and access to food and water *ad libitum*. All experiments were carried out in accordance with the European Community's Council Directive of September 22, 2010 (2010/63EU), and approved by the local animal care committee (AZ number 23 177-07/G 13-1-092). All animals used in this study were bred in-house and were allowed to acclimatize to the behavioral unit for at least 1 week.

### *Gadd45a-KO*
The *Gadd45a*-deficient mouse line was generated by Dr. Albert J. Fornace Jr. (National Cancer Institute, Maryland, USA) [13]. The first three exons of the *Gadd45a* gene were replaced by a phosphoglycerate kinase 1-neomycin resistance (neo) cassette. 19 amino acids are encoded by the remaining exon 4, but are not expressed [13]. *Gadd45a*-KO mice (in C57BL/6N background) grew to adulthood without obvious differences from their wild-type littermates. Genotyping of these animals was performed by polymerase chain reaction (PCR) as described [13].

### *Gadd45a overexpression*
A virally mediated recombination approach was established in order to overexpress *Gadd45a* in the neuronal population where this protein is endogenously expressed. Through a HpaI linker, the mouse *Gadd45a* open reading frame was fused at the 3′ end to hemagglutinin (HA) epitope tag-encoding sequences and cloned into an AAV expression cassette containing the 1.1 kb cytomegalovirus (CMV) immediate early enhancer/chicken β-actin hybrid promoter (CAG), a transcriptional terminator element flanked by two loxP sites ("Stop"), the woodchuck hepatitis virus post-transcriptional

regulatory element (WPRE), and the bovine growth hormone polyadenylation sequence (bGHpA) flanked by AAV2-inverted terminal repeats (construct termed as AAV-Stop-Gadd45a). The transcriptional terminator element ("Stop") was used to enable cell type-selective transgene expression [68,69]. Production of pseudo-typed AAV1/2 mosaic vectors and determination of genomic titers were performed as described [69]. Adult male mice (8–12 weeks old) were anesthetized by intraperitoneal injection of fentanyl (0.05 mg/kg), midazolam (5 mg/kg), and medetomidine (0.5 mg/kg), and positioned in a small animal stereotaxic frame (Kopf Instruments, California, USA). One microliter of the AAV-Stop-*Gadd45a* particles ($2.5 \times 10^{11}$ vector copies/ml) was injected bilaterally into the dorsal hippocampus (positions: −2.0 mm anteroposterior, ± 2.0 mm mediolateral, −2.0 mm dorsoventral of bregma) at a rate of 200 nl/min using a microprocessor controlled mini-pump with 34G beveled needles (World Precision Instruments, Florida, USA). AAV-Stop-*Gadd45a* vector injection into hippocampi of *Nex-Cre*$^{(+/−)}$ mice led to the excision of the transcriptional stop cassette in all Cre recombinase-positive cells [70], resulting in the transcriptional activation and the overexpression of virally encoded *Gadd45a*-HA in hippocampal glutamatergic neurons of the CA1-3 region, but not in dentate gyrus granule cells. As a control, AAV-Stop-*Gadd45a* was injected in *Nex-Cre*$^{(−/−)}$ mice, where the absence of Cre expression prevented the transcriptional activation of the construct.

### *Gadd45a*$^{K45E}$ point mutation

The substitution of lysine (K, coded by AAG) for glutamate (E, coded by GAG) at position 45 of the Gadd45α protein (K45E) was introduced in the original plasmid previously described (pAM/CBA-floxstop-HA-*Gadd45a*-WPRE-bGHpA) using the QuikChange II XL Site-Directed Mutagenesis Kit (Agilent Technologies, Germany) and the primers included in Appendix Table S2. Mutation was confirmed by sequencing, and the mutated plasmid was used for the production of AAV-*Gadd45a*$^{K45E}$ as previously described. Finally, the AAV-*Gadd45a*$^{K45E}$ was checked for adequate Gadd45α expression by qPCR and Western blot.

### Behavioral characterization

In order to avoid the possible confounding effects by multiple behavioral testing, two different batches of animals were separately investigated. The first batch (*Gadd45a*-WT, $n = 12$; *Gadd45a*-KO, $n = 16$) contained the following tests performed in the order described below.

#### Holeboard

The apparatus was made of white Plexiglas with an arena (40 × 40 × 30 cm) divided into 36 squares by black color strips. Four holes (diameter: 1.8 cm; depth: 7.5 cm) were present in the center where the animal could poke its nose, but their diameter did not allow the exploration with the whole body. The animal was always placed into the same corner of the arena and was allowed to explore during 5 min. The parameters recorded were total and internal (in the inner 16 squares) ambulation and frequency of head dipping into holes. This test provided independent measures of motor activity (internal and external ambulation) and directed exploration (head dipping). Animal was tracked using Ethovision XT software (Noldus Information Technology).

#### Elevated plus-maze

The apparatus consisted of a cross-shaped arena, elevated 100 cm above the floor, with two opposite open arms and two opposite enclosed arms. The floor of the arms was made of white plastic, 35 cm long and 6 cm wide, and connected by a central platform of 6 × 6 cm. Walls in black plastic of 20 cm height surrounded the enclosed arms. The animal was placed into the center of the apparatus, facing the enclosed arms, and was allowed to explore during 5 min. Frequency and duration of arm visits were measured, separately for open and closed arms. An arm was considered to be visited when the animal entered it with all four limbs. The percentage of entries and time spent in the open arms were calculated in relation to the total values for the open and enclosed arms using the following formula: % open arm = 100× open/(open+enclosed). Animals were tracked using Ethovision XT software (Noldus Information Technology).

#### Light/dark test

The apparatus in this assay consisted of an open white compartment (40 cm height × 26 cm length × 38.5 cm width) and one closed black compartment (40 cm height × 13 cm length × 38.5 cm width) connected by an entrance. The animal was allowed to move freely between the two compartments during 5 min. In the beginning of the experiment, the animal was placed directly in front of the entrance in the lit compartment. The total transitions between compartments, the percentage of time spent in the lit compartment, and the latency to the first entry into the lit compartment were measured. Animals were tracked manually.

#### Novel object recognition test

The test was performed in a white plastic open-field chamber (40 cm height × 40 cm length × 40 cm width). For habituation, the animal was placed into the empty open field and allowed to explore the box for 10 min once a day for 2 days. On day 3, two identical objects (O1 left and O1 right; two metal cubes of 4 cm height × 3 cm length × 5 cm width) were placed symmetrically 6–7 cm from the walls with a separation distance of 16–18 cm from each other. The mouse was placed into the box at an equal distance from both objects and was recorded on video for 10 min. After this first exposure to the objects, the mouse was returned to its home cage. 1 and 24 h later, the mouse was placed into the open field again. During these 10-min retention trials, it was exposed to the familiar object (O1), as well as to a novel object (O2 for the 1 h time point and O3 for the 24 h time point, respectively). The novel object O2 was a plastic billiard ball (5.72 cm in diameter) fixed on a metal plate (0.2 cm height), and O3 was a round glass flask (6 cm height × 3 cm wide) filled with sand and sealed with a black rubber plug. The familiar object was always positioned on the left side, while the new object was on the right side. Box and objects were cleaned with 70% ethanol after each trial to avoid olfactory cues. The total time that the animal spent exploring each of the two objects in training and retention phase was evaluated by an experimenter blind to the genotype. Object exploration was defined by orienting the nose directly to the object at a distance < 2 cm and/or touching the object with nose and whiskers. Time spent climbing and sitting on the object was not regarded as exploration as it rather represents a form of environment investigation. Animals were tracked manually. Using these

values, the discrimination index was calculated as a measurement of the recognition of the familiar object (discrimination index = (Time exploring O1/Time exploring both objects) × 100). Hence, a good performance is characterized by a reduction in the discrimination index over the subsequent trials.

### Forced-swim test

The paradigm was performed in a round glass beaker (18 cm in diameter; 45 cm height) filled with tap water at 25°C ± 0.5°C. The water level was approximately 30 cm to prevent the animal from touching the bottom of the glass or climbing off the beaker. The animal was carefully lowered into the water and recorded on video for 6 min. During the test, floating, struggling, and swimming were evaluated. Floating was defined by immobility of the animal with only minimal movements to keep body balance. Struggling was defined by rapid and strong movements performed in a vertical position, usually accompanied by scratching of the walls of the beaker. Swimming was then considered the time that the mouse did neither float nor swim. Animal was tracked manually. In this test, the natural aversion of animals to water and the unavoidable danger induced by this context exposure allowed us to interpret the actions of the mouse as an animal that tries to escape (those who swim faster and float less; classified as normal) or as an animal that does not try it and surrenders (slow swimming and more floating; classified as depressive-like symptom).

### Cued fear conditioning

Procedures and setups were used as previously described [71]. For conditioning, mouse was placed into the conditioning chamber (Med Associates; square, 15 × 20 cm, grid floor, cleaned with 1% acetic acid) and a house light (25 lx) turned on. After 3 min, a 20-s tone (80 dB, 9 kHz sine wave, 10-ms rising and falling time) was presented to the animals. The tone co-terminated with a 2-s scrambled electric foot shock of 0.6 mA. Mouse was returned to its home cage after 60 s. On day (d) 1, d2, d3, and d10 after the conditioning day, conditioned mouse was placed into a neutral, new environment (extinction context, custom-made Plexiglas cylinders, 15 cm diameter, with bedding, cleaned with 70% ethanol), and the house light (5 lx) was switched on. After 3 min, a 200-s continuous tone (same settings as in conditioning) was presented. Mouse was returned to its home cage 60 s after the end of the tone presentation. Animal was tracked using Ethovision XT software (Noldus Information Technology). Freezing (i.e., immobility, here defined as the absence of all non-respiratory movements) was scored with the Ethovision immobility filter set at 0.5% change in the pixels representing the mouse, with averaging over two consecutive frames (25 frames/s). Conditioned freezing, defined as freezing to the tone minus baseline freezing response of the same day, as well as baseline freezing before the tone started and freezing after the tone presentation ended were analyzed.

In the second phase of the behavioral characterization, a new batch of *Gadd45a*-deficient mice (*Gadd45a*-WT, $n = 11$; *Gadd45a*-KO, $n = 11$) and a batch of *Gadd45a*-overexpressing animals with their controls (*Gadd45a*-WT, $n = 10$; *Gadd45a*-overexpression, $n = 14$) were simultaneously investigated. All *Gadd45a*-overexpressing mice were validated by immunohistochemistry or qPCR. From the 14 animals initially injected with the AAV-Stop-*Gadd45a*, only two animals had to be excluded due to insufficient Gadd45α

overexpression. This second set of the behavioral characterization consisted of the following assays.

### Passive avoidance (PA)

This test was conducted in a box made of black and white Plexiglas, divided into two sections (15 cm height × 9.5 cm width × 16.5 cm length) (Ugo Basile, Italy) (Fig 1A). The chambers were separated by a flat-box partition consisting of an automatically operated sliding door at floor level. A light (24 V, 10 W, 90 lux) in the ceiling of the starting side was left on at all times, while the other side was kept in darkness. The lit compartment was white in color and the dark one in black. The floor was composed of stainless steel bars, 0.7 mm in diameter and 8 mm apart. In an acquisition trial, the mouse was initially placed into the illuminated compartment, and then, the door between the two compartments was opened 60 s later. When the mouse entered the dark non-illuminated compartment, the door closed automatically, and an electrical foot shock (0.3 mA in the experiment with *Gadd45a*-KO mice and 0.2 mA in the experiment with *Gadd45a*-overexpressing mice to avoid ceiling effects) was delivered through the stainless steel rods for 2 s. Three retention trials were performed by placing each mouse into the illuminated room 1 h, 24 h, and 7 days after the acquisition trial. The time taken for a mouse to enter the dark compartment after opening the door was defined as the latency time for the acquisition and retention trials. Animal was tracked manually. The latency prior to entering the dark compartment was recorded for up to 540 s. Afterward, if the animal did not visit the dark compartment, it was returned to its home cage.

### Water cross-maze

Procedures and setups were used as previously described [19]. The maze was made of 0.5-cm-thick and transparent Plexiglas and it had four arms forming a cross, with each arm 10 cm width, 50 cm length, and 30 cm height (Fig EV1F). For spatial orientation, four extra-maze visual cues were installed on the surrounding cubicle. The platform was located east; the starting point was either north or south. The arm opposite to the respective starting position was always blocked with a removable Plexiglas panel. The maze was filled with water at 22–23°C, until the platform was submerged 1 cm beneath the water surface. Animal was tracked using Ethovision XT software (Noldus Information Technology). The experiment was divided into two different phases, namely initial learning (platform always on east) and reversal learning (platform in west). During the initial learning, mouse was tested for 4 days with eight trials per day. The eight trials were performed in two sessions with four trials, allowing each mouse to rest for about 2 h between sessions. Each starting point was used twice per session, with the order of the starting points being randomized over the days. A trial ended when the mouse reached the platform and remained on it for at least 5 s or when the mouse did not find the platform within 30 s. In this case, the mouse was gently guided to the platform. The mouse had to stay on the platform for 30 s before it was removed, dried, and put back into its home cage. A heat lamp was used to warm the animals. Mice were trained in groups of six, resulting in an inter-trial interval (ITI) of 10 min within a session. The reversal learning phase was very similar except for the duration (2 days instead of four) and the location of the platform (west instead of east). Arm entries were manually scored. A trial was scored as accurate, if the animal swam from the starting arm directly into the goal

arm and climbed onto the platform. A trial was counted as non-accurate, if a mouse entered the arm without platform or if it visited an arm several times. Accuracy was defined as the percentage of accurate trials within each single session or day.

*Pain sensitivity tests*

In order to evaluate the pain sensitivity, two different tests, namely hot plate test (Ugo Basile, Italy) and foot shock threshold, were performed. In the first test, the mouse was placed on a surface heated at 48°C, and the latency to the first reaction to the heat (paw lifting, paw licking, or jumping) was scored. For the second test, the conditioning chamber of the cued fear conditioning apparatus was used. Animal was placed into the chamber, and a scrambled electric foot shock of rising intensity (starting from 0 mA, 0.015 mA/s) was applied. The shock was switched off as soon as the animals jumped or vocalized. The corresponding shock intensity was defined as pain threshold.

**In situ hybridization**

In order to localize the endogenous *Gadd45a* mRNA, *in situ* hybridization on frozen sections of unfixed adult brain was performed essentially as previously described [72]. The staining was carried out with chromogen nitro blue tetrazolium and 5-bromo-4-chloro-3′-indolyphosphate (NBT/BCIP). For this experiment, C57BL/6N mice ($n = 8$) were used as controls (same genetic background as the *Gadd45a* mutant line) and also *Gadd45a*-KO tissue ($n = 6$) in order to test the specificity of the riboprobe. Animals were anesthetized with isoflurane and euthanized by decapitation. Immediately afterward, brains were removed and placed in a brain matrix (World Precision Instruments, USA) where a thick coronal section containing the entire hippocampus was separated from the rest of the brain. This section was immediately frozen and further cut in 10- to 12-μm sections. On the first day of ISH, sections were thawed for 2 h, post-fixed in 4% paraformaldehyde (PFA, Merck) in 0.1 M Na-phosphate buffer, pH 7.4, acetylated for 10 min, and hybridized with digoxigenin-labeled riboprobes overnight (o/n) at 65°C. The next day, sections were washed for 1 h in 50% formamide in 1× saline sodium citrate buffer (SSC) and 0.1% Tween-20 at 65°C. Afterward, they were blocked for 2 h and incubated with 1:1,500 anti-DIG-AP antibody (Roche) in 1% Boehringer Blocking Reagent (Roche, Germany), 10% heat-inactivated goat serum in *maleic acid* buffer containing Tween-20 (MABT; 0.1 M maleic acid, 0.15 M NaCl, pH 7.5, and 0.1% Tween-20) o/n at 4°C. Sections were 3 × 15 min washed in MABT and stained with 1 μl/ml NBT/BCIP (Roche, Germany) in alkaline phosphatase buffer (NTMT; 0.1 M Tris–HCl, pH 7.5, 0.15 M NaCl, 0.05 M MgCl₂, 0.1% Tween-20). *Gadd45a* antisense and sense riboprobes were synthesized with digoxigenin RNA labeling mix (Roche, Germany) according to manufacturer's instructions, using an amplified cDNA fragment (position 11-215 of NM_007836.1) as template. Images were taken with a DM2500 microscope (Leica, Germany) and processed with Adobe Photoshop software.

**Quantitative polymerase chain reaction (qPCR)**

Mice ($n = 6$–8; per experimental group) were anesthetized with isoflurane and euthanized by decapitation. Immediately afterward,

brains were removed and placed in the same brain matrix mentioned above. 2- to 3-mm sections were extracted from the brain matrix corresponding to the regions where hippocampus and control areas (prefrontal cortex and cerebellum) were located according to the mouse brain atlas [73]. Sections were dissected with a scalpel under the binocular in a small petri dish filled with ice and covered with a circular paper embedded in 400 μl of diethylpyrocarbonate (DEPC)-water to avoid RNA degradation. Tissues were immediately frozen and stored at −80°C. Samples were collected either under basal conditions or at given time points after memory acquisition in the PA paradigm. Frozen samples were processed subsequently in order to extract total RNA. Samples were homogenized using the TissueLyser (Qiagen, Netherlands) according to the manufacturer guidelines. Total RNA was extracted using the RNeasy kit (Qiagen, Netherlands) and finally eluted in 40 μl of RNase-free water. RNA content was quantified using the NanoDrop (Thermo Scientific, Germany), and samples were stored at −80°C. On the next day, samples were thawed, and equal amounts of RNA from all samples (500 ng or 1 μg depending on the experiment) were taken for the conversion to complementary DNA (cDNA). At this point, two different strategies were followed for the priming in the reverse transcription. (i) Random primers included in the high-capacity cDNA reverse transcription kit (Life Technologies, Germany) were used. (ii) In a number of experiments aiming at the evaluation of polyadenylated RNA, the approach consisted on using oligo(dT) primers that were included in the SuperScript III First-Strand Synthesis System for RT-PCR kit (Life Technologies, Germany). In both cases, the resulting cDNA was diluted 1:10 in RNase-free water and stored at −80°C. Finally, the cDNA was amplified in two different manners. On the one hand, by using the commercial TaqMan assays detailed in Appendix Table S2 (Applied Biosystems) plus TaqMan gene expression master mix (Life Technologies, Germany). On the other hand, for all the experiments analyzing *Grin2a* 3′UTR, specific primers for the distal (3′ end) and proximal part (5′ end) of the 3′UTR were designed with Prime3 software [74]. These primers (Appendix Table S2) were used in combination with the PowerUp SYBR Green Master Mix (Life Technologies, Germany). Assays were carried out with an ABI7300 real-time PCR cycler (Applied Biosystems). Reactions were performed in duplicates using *glucuronidase beta* (*Gusb*, for TaqMan assays) and/or *transferrin receptor* (*Tfrc, for SYBR green assays*) as reference gene. Data analysis was done with the 7300 system SDS software (Applied Biosystems). Ct values were transformed to mRNA expression using the following formula: $2^{-(\text{Ct.GOI} - \text{Ct.HKG})}$, where GOI = gene of interest and HKG = housekeeping gene. All values were then normalized to the specific control group for each experiment.

**Immunohistochemistry**

For the detection and validation of the viral delivery of *Gadd45a*, immunostainings against HA (fused at the C terminus of the *Gadd45a* in AAV construct) were performed as previously described [70]. Mice were perfused with 4% PFA in 0.1 M phosphate-buffered saline (PBS, pH 7.4). Brains were removed from the skull, post-fixed o/n, and incubated in 30% sucrose (in PBS) for cryoprotection. Brains were cut (40 μm) in the coronal plane on a cryostat. Free-floating sections were rinsed in PBS with 0.2% Triton X-100

(PBS-T). Non-specific immunoreactivity was suppressed by incubating the sections in a cocktail of 4% normal goat serum (Sigma, Germany) in PBS-T for 15 min at 22–24°C. Afterward, sections were incubated o/n with a mouse anti-HA antibody (Covance, USA; 1:1,000). Sections were washed and incubated for 1 h with Alexa 488-conjugated goat IgG (1:1,000, Invitrogen, USA). Before the third wash in PBS, sections were counterstained with the nuclear dye 4′,6-diamidino-2-phenylindole (DAPI) for 5 min. Sections were then transferred onto glass slides and cover-slipped with Mowiol mounting medium. Fluorescence was visualized using either a Leica DM5500 microscope (Leica, Germany) or a Leica TCS SP5 confocal microscope.

### Electrophysiological recordings

In the experiments with *Gadd45a*-KO mice, 11-12 animals per experimental group were used. *Gadd45a*-overexpressing animals (group size of 6 mice per genotype) were tested 8 weeks after brain surgery. Age-matched male mice (between the 7th and 15th post-natal week) were anesthetized by isoflurane, killed by cervical dislocation, and the brain was immediately removed from the skull and immersed in ice-cold carbogen-equilibrated ($5\%CO_2/95\%O_2$) artificial cerebrospinal fluid (ACSF) containing (in mmol/l) NaCl 126, $NaH_2PO_4$ 1.25, $MgCl_2$ 1, $CaCl_2$ 2, KCl 2.5, glucose 10, and $NaHCO_3$ 26. After 2–3 min of incubation, transverse slices of 400 μm thickness containing the amygdala or the hippocampus were prepared. The slices were transferred to an interface-type recording chamber where they were superfused with ACSF at a rate of 1–2 ml/min. Extracellular field potentials (FPs) were recorded with glass electrodes filled with ACSF placed in the lateral nucleus of the amygdala (LA) or in stratum radiatum of the CA1 region. Synaptic afferents were stimulated via a bipolar tungsten electrode placed in the cortical inputs to the amygdala or to the Schaffer collaterals in the CA3 region. LTP was induced by a series of 100-μs voltage pulses at half-maximal response intensity delivered in four trains of 100 pulses at 100 Hz with an inter-train interval of 1 s. FP responses were recorded for 15 min before and 85–90 min after LTP induction. For analysis, the slope between 10 and 90% of the FP responses was calculated using an Excel-Script. Experiments were discarded from analysis if they did not show stable FP results or if unexpected drops in excitability or slope occurred during the recording interval. All measurements and analyses were made without knowledge of the genotype. Statistical comparisons between test and control values were performed using paired Student's *t*-test, comparisons between groups by unpaired Student's *t*-test. Differences in incidence rates were tested with Fisher's exact test (Systat 11, SPSS).

### RNA sequencing

RNA was extracted from hippocampi from six *Gadd45a*-WT and six *Gadd45a*-KO mice, which were trained in the PA and killed after 1 h. Sample collection and RNA extraction were performed as described above for qPCR. Nanodrop measurements for RNA content revealed purity of the RNA extracted (ratio of absorbance at 260/280 always above 2.0; generally accepted as pure RNA isolation) and low contamination with other molecules contained in the extraction buffers (ratio of absorbance at 260/230 usually on the range of 2.0–2.2). RNA was further tested for quality control regarding the RNA integrity number (RIN). This parameter was measured by Agilent 2100 Bioanalyzer, obtaining a range of 8.8–9.4 for all samples, which was considered sufficient. The next-generation sequencing (NGS) libraries were prepared using the TruSeq Stranded mRNA Library Prep Kit (Illumina, California, USA) following manufacturer's instructions. Sequencing was performed on HiSeq 2000 sequencer in SR51 mode. The quality of the raw sequence reads (around 60 million per sample) was very good as assessed by FastQC (https://www.bioinformatics.babraham.ac.uk/projects/fastqc/). Reads were mapped to the mouse reference genome (Illumina iGenomes reference, Ensembl NCBIM37/mm9 build) using the splice-aware mapper TopHat v.1.4.1 (https://ccb.jhu.edu/software/tophat/index.shtml) with parameters "-g 5 –G" (up to five best alignments allowed per read and usage of a GTF gene annotation file from iGenomes, Ensembl release 66). Quality assessment of the mapped data was performed with RSeQC (http://rseqc.sourceforge.net/). Metagene profiles of transcript coverage were made using ngs.plot (https://github.com/shenlab-sinai/ngsplot) with parameters "–G mm9 –R genebody –F rnaseq". The read counts per gene were summarized using the tool *htseq-count* from the package HTSeq v.0.5.4p3 (http://www-huber.embl.de/HTSeq/doc/count.html) and stranded option "-s reverse". Differential gene expression analysis was performed with the BioConductor package DESeq v.1.16.0 (http://bioconductor.org/packages/release/bioc/html/DESeq.html) following the standard routines recommended by the software authors [75]. Using a cut-off of 1% false discovery rate (FDR), the DESeq analysis revealed 407 genes with apparent differential expression (66 with a two-fold change threshold), almost all of them being down-regulated. The list of deregulated genes was annotated through the Mouse Genome Informatics (MGI) database (http://www.informatics.jax.org/batch). For browser visualization, the mapped data in BAM format were converted into reads-per-million (RPM)-normalized bedGraph and bigWig coverage tracks using PICARD v.1.56 (https://broadinstitute.github.io/picard/), BEDTools v.2.16.2 (http://bedtools.readthedocs.io/), SAMtools v.0.1.18 (http://samtools.sourceforge.net/), and the script *bedGraphToBigWig* (http://hgdownload.soe.ucsc.edu/admin/exe/). The normalized bigWig tracks were displayed on the UCSC mm9 genome browser (http://genome.ucsc.edu/cgi-bin/hgGateway), and screenshots of regions of interest were taken using one representative replicate per genotype depicted at the same RPM scale. Gene ontology (GO) enrichment analysis was carried out with DAVID Bioinformatics Resources 6.8 (https://david.ncifcrf.gov/) with a background list of all genes showing detectable expression in the RNA-seq experiment. Read summarization at the exon level was carried out using Subread v.1.5.1 (http://subread.sourceforge.net/) with options "featureCounts -f -s 2 -O -F GTF" and using the above-mentioned GTF gene annotation supplemented with four "pseudo-exons" representing the distal 1 kb regions of the extended 3′UTRs of the four genes of interest *Grin2a*, *Grin2b*, *Grm5*, and *Kcnq3*. Exon-level differential analysis was then performed using DESeq2 v.1.18.1 at 1% FDR (https://bioconductor.org/packages/release/bioc/html/DESeq2.html). RPM-normalized exonic and intronic coverage for the four investigated genes was calculated using BEDTools v.2.25.0, SAMtools v.1.5, and Subread v.1.5.1. The RNA-seq data have been deposited in the NCBI GEO repository (http://www.ncbi.nlm.nih.gov/geo) under accession number GSE100923.

## Synaptosomal preparation and Western blotting

Sample collection was performed as described above for qPCR. Hippocampi ($n$ = 13-14/experimental group) were frozen immediately after brain dissection and stored at −80°C. Synaptosomal preparation was carried out as described previously [76,77]. Whole hippocampus was homogenized in a glass grinder with 1.5 ml of TVEP buffer (10 mM Tris–HCl pH = 7.4, 5 mM NaF, 1 mM $Na_3VO_4$, 1 mM EDTA, 1 mM EGTA) containing 320 mM of sucrose. The homogenate was centrifuged (10 min, 1,000 × $g$ at 4°C), and the pellet containing nuclei and large debris was discarded. An aliquot of the supernatant containing the total fraction (110 µl) was removed and stored at −80°C for later analysis. The rest of the supernatant was further centrifuged (15 min, 15,000 × $g$ at 4°C) in order to obtain crude synaptosomes (pellet). For mRNA analysis of synaptosomal fractions, samples were processed at this step for total RNA extraction (RNeasy kit, Qiagen, Netherlands) as described above in the qPCR section, although with a reduction in the elution volume from 40 to 20 µl. Otherwise, pellet was then re-suspended in 1.5 ml of TVEP buffer containing 35.6 mM of sucrose and placed on ice for 30 min. This step led to the hypoosmotic lysis of the synaptosomes in order to release synaptic vesicles and other organelles. Finally, the last centrifugation (20 min, 20,000 × $g$ at 4°C) yielded a pellet enriched in plasma membrane proteins (synaptosomal fraction), which was re-suspended in 200 µl of TVEP buffer and stored at −80°C. Total and synaptosomal fractions were analyzed for protein concentration using the bicinchoninic acid assay (Pierce BCA Protein Assay Kit; Thermo Scientific, Germany). For gel electrophoresis and Western blotting, total and synaptosomal fractions were analyzed separately. Each gel was loaded with equal amounts of total (5 µg) or synaptosomal (3 µg) proteins. Samples were mixed with 8–10 µl of 3× loading buffer (200 mM Tris–HCl pH 6.8, 8% sodium dodecyl sulfate (SDS), 25% glycerol, 10% β-mercaptoethanol, and 0.02% (w/v) bromophenol blue) to a total volume of 20 µl and incubated for 10 min at 95°C. Afterward, two samples/experimental group (for a total of eight samples/gel) were loaded in 7.5% polyacrylamide gel (PAGE) and run at a constant voltage of 90 V for 1 h in Mini-PROTEAN® cassettes connected to a PowerPAC3000® electric unit (BIORAD, Germany). Subsequently, protein transfer to the nitrocellulose membranes was conducted for 1.5 h at a constant current of 300 mA in the same cassettes. Then, membranes were cut into upper and lower sections for the separate detection of NR2A and loading control, respectively (tubulin as a control for the total fraction and post-synaptic density protein 95 (PSD95) for the synaptosomal fraction). Membranes were blocked for unspecific signal with 5% non-fat dry milk (NFDM) for 1 h. Afterward, primary antibodies for NR2A (MAB5530, Merck-Millipore, 1:1,000), mGluR5 (AB5675, Merck-Millipore, 1:2,000), tubulin (T9026, SIGMA, 1:500), synaptotagmin (sc-136089, Santa Cruz, 1:500), and PSD95 (61495, BD Biosciences, 1:250) were incubated overnight in 5% NFDM at 4°C. On the next day, membranes were washed and incubated with horseradish peroxidase (HRP)-conjugated mouse secondary antibody (Dianova, Hamburg, Germany; 1:5,000). Signal was visualized using the ECL detection system according to manufacturer instructions (GE Healthcare Life Sciences). Visualization and quantification of the bands were done with Fusion and Bio-1D software, respectively (Vilber Lourmat). Finally, the ratio between the signal of NR2A/ mGluR5and the respective loading control (tubulin for the total fraction and synaptotagmin for the synaptosomal) was calculated for each sample. These ratios were normalized to the average of the ratios of the control groups (no PA; set to 100%) for each fraction and genotype.

## Nuclear/cytosolic fractionation

Sample collection was performed as described above for qPCR. Hippocampi from *Gadd45a*-WT and *Gadd45a*-KO ($n$ = 2 for both groups) were frozen immediately after brain dissection and stored at −80°C. For the separation of the main cellular compartments, the nuclear/cytosol fractionation kit (Biovision, USA) was used according to the manufacturer guidelines. Validation of the enrichment of proteins in the two fractions was done by Western blot as previously described. Primary antibodies for laminin (nuclear protein; Ab16048, Abcam, 1:1,000) and microtubule-associated protein 2 (MAP2, cytosolic protein; MAB3418, Merck-Millipore, 1:1,000) were used in combination with (HRP)-conjugated rabbit and mouse secondary antibodies, respectively.

## Expression and purification of Gadd45α-FLAG and GFP-FLAG proteins

Human embryonic kidney (HEK) 293 cells cultured in 15-cm dishes were transfected at 60% confluency with 10 µg of $pCS2^+$ plasmid-encoding human Gadd45α-FLAG (94.5% identity with mouse Gadd45α; Fig EV5E) or *GFP*-FLAG protein using TransIT reagent (Mirus). HEK293 cells were harvested 48 h post-transfection and lysed in 1 ml of lysis buffer (50 mM Tris–HCl (pH 7.5), 300 mM KCl, 5 mM EDTA, 1 mM DTT, 0.5% Triton X-100, and 50 µl of Roche proteinase inhibitors); after homogenization and brief sonication, the cell lysates were incubated for 30 min at RT in the presence of 4 µg/ml of RNase A (Fermentas) and 400 U/ml of benzonase (Merck-Millipore). Cell lysates were centrifuged for 10 min, 13,000 × $g$ at 4°C. Supernatants were cleared using 100 µl of protein G agarose beads for 1 h at 4°C. After centrifugation (2,000 × $g$, 5 min), supernatants were incubated with 50 µl of M2-FLAG-agarose BSA-pre-blocked beads (Sigma) for overnight at 4°C. Beads were washed extensively, and bead-bound proteins were analyzed on Coomassie blue-stained 12% SDS–PAGE gel and quantified using BSA as a standard (Fig EV5F).

## Pulldown of Gadd45α-FLAG-bound RNAs in mouse hippocampus

Mouse hippocampi were dissected and frozen on dry ice as previously described for qPCR. Frozen samples were lysed using tissue homogenizer in 600 µl of lysis buffer (50 mM HEPES-KOH (pH 7.5), 100 mM KCl, 5 mM EDTA, 1 mM DTT, 0.5% Triton X-100, 1 mM PMSF, Roche proteinase inhibitors, and 0.2 U/µl Promega RNase inhibitors) and incubated on ice for 30 min. Tissue lysates were centrifuged at 13,000 × $g$ for 10 min at 4°C. Supernatants were incubated with 20 µl of protein G agarose beads for 1 h at 4°C. Lysates were recovered by centrifugation (2,000 × $g$, 5 min) and incubated overnight with 250 ng of agarose bead-bound Gadd45α-FLAG or GFP-FLAG at 4°C. Beads were then washed five times for 10 min using a rotating device with 1 ml of lysis buffer and one time with lysis buffer containing 1 M urea. Protein-bound RNAs were eluted

in 100 μl of 0.5% SDS and 20 μg of proteinase K incubated at 37°C for 15 min. RNA was extracted from eluates using TRIzol (Life Technologies) and precipitated with isopropanol in the presence of 20 μg of glycogen. cDNA was produced by random hexamers primers and Superscript II kit, following the manufacturer instructions (Life Technologies, Germany). qPCR was carried using TaqMan primers (Appendix Table S2) on a Roche Light Cycler. Percentage of input recovery was calculated using the Ct values of amplification (qPCR) for the input fraction (10% pre-incubated lysates; CI) and the Gadd45α-FLAG- or GFP-FLAG-bound (CG). These values were then transformed according to the following formula: % Input recovery = $100 \times 2^{((CI-3,32)-CG)}$.

### Statistics

For multiple comparisons, data were analyzed using one- or two-way analysis of variance (ANOVA) followed by Bonferroni post hoc test and Tukey's multiple comparisons test when necessary (GraphPad, San Diego, CA, USA). In some cases, comparisons were made by using Student's *t*-test analysis (paired and unpaired), repeated measures ANOVA and/or Fisher exact test. The exact statistical analysis used for each experiment is included in the figure legends. Differences were considered statistically significant if $P < 0.05$. Data were presented as mean ± SEM.

Expanded View for this article is available online.

### Acknowledgements

We gratefully acknowledge the support of the Genomics Core Facility at the IMB. This work was supported by the German Research Foundation DFG (CRC1080; subprojects granted to BL, CN, and HJL).

### Author contributions

Conception and/or design of the investigation: BL, CN, AAR, and EK; methodology: BL, CN, AAR, and EK; data collection: AAR, SS, KA, AS, MG, SG, CB, LP, WHG, RJ, CM, and AC; data analysis and interpretation: AAR, EK, SS, WK, and HJL; writing–original draft: AAR; writing–review and editing: BL, CN, AAR, and EK; supervision: BL and CN; funding acquisition: AAR, HJL, CN, and BL.

### Conflict of interest

The authors declare that they have no conflict of interest.

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
