## [Review Process File · EMBO Reports]

Gadd45a modulates aversive learning through post-transcriptional regulation of memory-related mRNAs

Alejandro Aparisi Rey, Emil Karaulanov, Salim Sharopov, Khelifa Arab, Andrea Schäfer, Mathias Gierl, Stephan Guggenhuber, Caroline Brandes, Luigi Pennella, Wolfram H. Gruhn, Ruth Jelinek, Christina Maul, Andrea Conrad, Werner Kilb, Heiko J. Luhmann, Christof Niehrs & Beat Lutz

Review timeline:

Submission date:	28 February 2018
Editorial Decision:	4 April 2018
Revision received:	21 December 2018
Editorial Decision:	25 January 2019
Revision received:	22 February 2019
Accepted:	7 March 2019

Editor: Esther Schnapp

Transaction Report:

1st Editorial Decision

4 April 2018

Thank you for the transfer of your manuscript to EMBO reports. We have now received the full set of referee reports that is copied below.

As you will see, the referees acknowledge that the findings are potentially interesting. However, they also point out that the mechanistic model should be strengthened, that the conclusions need to be toned down and that better statistical analyses need to be performed. I think all referee comments make sense and should therefore be addressed.

We would thus like to invite you to revise your manuscript with the understanding that the referee concerns must be fully addressed and their suggestions taken on board. Please address all referee concerns in a complete point-by-point response. Acceptance of the manuscript will depend on a positive outcome of a second round of review. It is EMBO reports policy to allow a single round of revision only and acceptance or rejection of the manuscript will therefore depend on the completeness of your responses included in the next, final version of the manuscript.

Revised manuscripts should be submitted within three months of a request for revision; they will otherwise be treated as new submissions. Please contact us if a 3-months time frame is not sufficient for the revisions so that we can discuss this further. Given the 6 main figures, I suggest that you layout your manuscript as a full-length article with separate results and discussion sections. Please change the reference style to the numbered EMBO reports style that can be found in EndNote.

Supplementary figures, tables and movies can be provided as Expanded View (EV) files, and we can offer a maximum of 5 EV figures per manuscript. EV figures are embedded in the main manuscript text and expand when clicked in the html version. Additional supplementary figures will need to be included in an Appendix file. Tables can either be provided as regular tables, as EV tables or as Datasets. Please see our guide to authors for more information.

Regarding data quantification, please specify the number "n" for how many independent experiments were performed, the bars and error bars (e.g. SEM, SD) and the test used to calculate p-values in the respective

figure legends. This information must be provided in the figure legends. Please also include scale bars in all microscopy images.

We now strongly encourage the publication of original source data with the aim of making primary data more accessible and transparent to the reader. The source data will be published in a separate source data file online along with the accepted manuscript and will be linked to the relevant figure. If you would like to use this opportunity, please submit the source data (for example scans of entire gels or blots, data points of graphs in an excel sheet, additional images, etc.) of your key experiments together with the revised manuscript. Please include size markers for scans of entire gels, label the scans with figure and panel number, and send one PDF file per figure.

- a complete author checklist, which you can download from our author guidelines (<http://embor.embopress.org/authorguide#revision>). Please insert page numbers in the checklist to indicate where in the manuscript the requested information can be found. The completed author checklist will also be part of the RPF (see below).
- a letter detailing your responses to the referee comments in Word format (.doc)
- a Microsoft Word file (.doc) of the revised manuscript text
- editable TIFF or EPS-formatted figure files in high resolution. In order to avoid delays later in the process, please read our figure guidelines before preparing your manuscript figures at: http://www.embopress.org/sites/default/files/EMBOPress_Figure_Guidelines_061115.pdf

I look forward to seeing a revised version of your manuscript when it is ready. Please let me know if you have questions or comments regarding the revision.

REFEREE REPORTS

Referee #1:

In the manuscript by Rey et al. entitled "Gadd45a regulates memory consolidation via stabilization of mRNAs of synaptic plasticity proteins", the authors explore the role of the Gadd45a protein in memory through studies of an established Gadd45a knockout mouse. Through a transcriptomic analysis, the authors identify candidate targets of Gadd45a and they propose a model that Gadd45a regulates the stability of target RNAs that are critical for neuronal function. There are some aspects of the work that require more evidence to support the model. Mechanistic studies are challenging *in vivo*, but some experimental approaches could provide support for the model.

The data used to argue for a change in RNA stability is not sufficient to support this model.

The authors employ a previously generated and characterized Gadd45a knockout mouse. However, as much of the data presented relies on this mouse, the mouse should be described in some detail in the Introduction. No information is provided here about the nature of the knockout with only some minor comments about the characterized phenotypes. While characters may be limited, there is a need for at least some introduction to this mouse.

Specific Comments:

Figure 1 presents the initial evidence that the Gadd45a mouse has defects in memory consolidation. The order of Figure is rather odd. The finding that Gadd45a transcript is present in the CA 1-3 regions really provides the rationale to assess hippocampal functions. Thus, current Figure 1C, which also provides

evidence that the knockout mouse actually lacks Gadd45a transcript in these regions of the brain seems like the logical Figure 1A. The image of the apparatus used for the experiment currently shown in Figure 1A is not informative and the Figure Legend does not add to the value of the image. If the authors wish to illustrate this behavioral paradigm, they could employ a schematic that illustrates the apparatus and depicts the experiment. The experiment in current Figure 1D is only mentioned in passing with no comparison of Gadd45a levels to Gadd45b or Gadd45g, despite the fact that these are the data presented in this figure. The relative RNA comparison shown in Figure 1E is not that valuable as RNA levels do not dictate protein levels, which is really the valuable comparison. These data do raise the question of how the function of Gadd45a is coordinated with or complemented by the function of Gadd45b and/or Gadd45g.

Figure 2 analyzes the consequences of overexpression of Gadd45a. The authors missed an opportunity to perform these experiments with a version of Gadd45a that should not binding RNA; however, the experiments presented still argue that overexpression of Gadd45a is sufficient to enhance performance in the memory consolidation assay employed. This finding is surprising, which is why having a control that uses a version of Gadd45a that cannot bind to RNA would have been ideal. In fact, amino acid substitutions that decrease the interaction of Gadd45a with RNA have been created and characterized (Sytnikova et al., PLOS One, 2011).

Figure 4 presents RNA-Seq data comparing the Control and Gadd45a knockout mouse. The authors argue that the transcripts altered in the Gadd45a samples contain an extended 3'UTR. From the presentation of the data, it is not clear whether the authors compared this sample of transcripts to a control set from the same samples. These extended 3'UTR could be diagnostic of the tissue used for the study rather than specific to loss of Gadd45a. The data presented in Figure 4E is not sufficient to argue for a difference in the 3'UTR. There could be a bigger difference in the low level reads at the start of the 3'UTR as compared to the many reads at the end of the 3'UTR.

Figure 5 examines the Grin2a transcript as a target of Gadd45a. The argument that a difference in random-primed cDNA as compared to oligo(dT)-primed cDNA supports a model of altered stability. A surrogate measure for transcription would be to examine pre-mRNA, which is rapidly processed and thus an acceptable measure of transcription. These results could be indicative of alternative cleavage/polyadenylation rather than the authors' conclusion regarding RNA stability. Comparing the data presented in Figure 4E and Figure 5D/E is confusing because the difference in the control levels of 3'UTR transcript levels compared to the Gadd45a-KO is lost in the relative mRNA levels presented. These data should be plotted on the same graph to allow comparison.

Figure 6 presents mechanistic insight into the difference in the level of the Grin2a protein product (NR2A) in the synaptosomal fraction. With these fractions in hand, the authors could address whether the levels of Gadd45a transcript are locally altered. The RNA binding experiment is not very robust. With this experiment, the interaction could be indirect- is the percent input recovery really 0.25%?

There are some aspects of the model that do not make sense. The Gadd45a protein is localized to nuclear speckles, which is typically a site of RNA processing. The authors do not present evidence that Gadd45a is located in the cytoplasm (or outside the nucleus). Most typically control of RNA stability occurs in the cytoplasm and the argument here might be that there are local changes to impact local NR2A levels. The authors need to address these points in a mechanistic model. As presented, there is not sufficient evidence to support the authors' model. With the synaptosomal fractions, the authors have the potential to directly address some of the mechanistic questions raised such as whether Gadd45a can be detected in these fractions.

Minor points:

The Table shown in Figure 4B is very difficult to read even when viewed in a magnified PDF image. The number of significant figures shown for the fold enrichment and the p-values shown seems rather excessive.

Referee #2:

Rey et al nicely investigated roles of Gadd45a in learning and memory and synaptic plasticity using knockout mice and overexpression mice. The authors have shown that Gadd45a KO mice showed impaired long-term memories but showed normal short-term memories. These KO mice also showed impaired hippocampal LTP. Conversely, Gadd45a overexpression improved long-term memory. These observations suggest that Gadd45a positively controls memory consolidation and LTP. Furthermore, the authors found that Gadd45a deficiency destabilizes mRNAs for NR2A and mGluR5 that is bound with Gadd45a, leading to reductions of these protein levels. The authors extensively performed behavioral, electrophysiological and biochemical experiments and showed important findings. The results are so clear. I have only minor

concerns as follows.

1) I recommend that the authors discuss about molecular mechanism of impairments in memory consolidation and LTP by Gadd45a KO. Although post-transcriptional roles of Gadd45a in mRNAs for synaptic proteins are well discussed in the Discussion session, they did not well discuss about how Gadd45a KO and overexpression impaired or improved, respectively, LTP and memory. Do the authors hypothesize that reduction of NR2A and mGluR5 is main reason leading to impaired LTP and memory?

2) P4 (Result of Fig 1C). "The signal localization suggests that Gadd45a mRNA is restricted to the cell body of principal glutamatergic neurons" is an overstatement. The authors did not show pictures with higher magnification and in situ hybridization using other marker probes.

3) From the results of Figure 5 and sFig.9, it is a bit difficult to understand (suppose) existence forms of Gadd45a mRNA. I am curious for results of northern blotting using specific probes.

Referee #3:

Rey et al study the role of the RNA binding protein Gadd45a in memory function and report that loss of Gadd45a in mice leads to impaired synaptic plasticity and learning behavior, while its over-expression improves specific forms of memory consolidation. The authors provide evidence that Gadd45a is involved in the turn over of mRNA coding for synaptic proteins. This is a very interesting study on a timely topic. The link of Gadd45a to the stability of mRNA coding of synaptic proteins is particularly interesting, although this part is less developed than the rest of the study and - at present - remains somewhat speculative. I understand, however, that the many experiments that come to mind are beyond the scope of the manuscript and suggest that many issues can be solved via rewording and I think the study In conclusion the presented data are very intriguing and the study should be published in EMBO reports.

In the following some suggestions/questions:

1.

The title somewhat overstates the data. While the authors convincingly show that Gadd45b regulates memory consolidation, the data to support the view that this finding critically depends on mRNA stabilization is interesting but would need more work to conclusively demonstrate this link. Similarly the first sentence of the discussions reads "Our study demonstrates for the first time a neural function for Gadd45a specifically in the consolidation of aversive memory and LTP, thereby contributing to the understanding of memory processing and uncovering a novel post-transcriptional mechanism involving extended 3'UTRs." Such statements should be toned down in my view.

2.

I think the authors should make clear that the regulation of mRNA stability is one potential mechanisms by which Gadd45a regulated memory formation. In fact, RNA stability itself has not been tested directly nor has the concept been explained (i.e. what factors degrade RNAs, how can that be prevented, how is it initiated). Moreover, Gadd45a has been described to be involved in other pathways as well, notably DNA demethylation and genomic stability. It is somewhat surprising that the authors did not test DNA-methylation. Is there a specific reason for this? At least such alternatives should be mentioned in the discussion.

3.

Does Gadd45a bind to the selected target mRNAs?

4.

Fig. 5D: the increase of proximal signal would imply a higher abundance of shorter transcripts, whereas longer transcripts would remain at the same concentration. The authors however conclude: "we observed that on the one hand the distal part of the Grin2a 3' UTR was not significantly different between genotypes and experimental groups, indicating absence of differences at the transcriptional rate" which is not evident from the data. Furthermore: according to their mRNA stability hypothesis, transcripts from KO mice should be degraded faster, thus for getting the same abundance of transcripts the transcriptional rate of KO mice would have to be actually higher. This should be discussed.

5.

Fig. 6A shows "activity-dependent NR2A membrane insertion (Fig 6A). Of note, this process has been previously shown to depend on local translation of NR2A, which is in turn controlled by the 3'UTR of the Grin2a transcript."

Is Gadd45a present in synaptosomes? As the authors mention, local translation is more likely to account for the increase of NR2A, which would imply the presence of a Gadd45a bound and stabilized Grin2a transcripts at the synapse.

In the discussion the authors talk about the potential of Gadd45a mediating the "long-distance transport of RNAs to different subcellular locations (Tiedge, 2006)". Maybe in the knockout mouse that transport failed and thus the Grin2a never arrives at the synapse?

6.

Fig 4D&E establish very clearly that in KO mice much less Grin2a transcripts contain exons than in the WT. But in figure 5A the authors see the same quantity of Grin2a RNA between both groups when using primers against the very same exons that they have shown to be way less abundant (p.7: "To validate these findings, we performed qPCR for Grin2a (Taqman assay covering exons 3-4) on RNA extracted from hippocampus after 1 hour following PA training, using cDNA reverse-transcribed either with random primers or with oligo(dT) primers (priming at the 3' end of polyadenylated RNAs)"). This should be clarified.

7.

p. 6: "As a result, the read coverage differences of Grin2a exons between Gadd45a-WT and Gadd45a-KO are clearly observable throughout the protein coding sequence, at the beginning of the 3'UTR but not at the 3' end of the extended (non-canonical) 3'UTR. This observation suggested an alternative interpretation of the mRNA-seq results, namely a Gadd45a-dependent post-transcriptional mechanism that does not affect the total amount of initially transcribed mRNA, but rather mRNA stability."

The authors neither provide literature that would point in the mRNA stability direction, nor do they exclude other possibilities by reasoning and/or experiments.

8.

The authors talk about a significant exon gradient (Fig. 4E description "Detailed view of the extended 3'UTR of Grin2a showing a significant 5' < 3' gradient in read coverage. Note that the read coverage is similar between genotypes at the actual hippocampal 3'UTR end (lower panel at right side), but differs significantly in the canonical annotated 3'UTR and over all exons (e.g. exons 4-8)."), but do not provide statistical prove for that significance. R packs like DEXseq could do such analyses. Moreover inspection by eye does not reveal a clear gradient between 5' to 3' but rather a global dampening of exon usage. Sentences like "However, the consistently reduced steepness of this coverage gradient in KO as compared to WT is indicative of increased transcript instability associated with Gadd45a deficiency." (p.6) are not well grounded. If the mRNA stability hypothesis was true, then in line with the result that the 3' UTR read coverage is the same between WT and KO, the only possibility of mRNA decay in Gadd45a-KO mice would be an RNase activity from 5' → 3' (which is one of the known mechanisms for mRNA decay. As this would happen at different timepoints from cell to cell and transcript to transcript one would expect indeed a gradient in the exonal coverage: more transcripts would be captured with at least a few exons fully covered towards the 3' end, whereas almost no transcript would be seen with reads over the first exons (since they would then be the first ones getting decayed). Thus, it would be crucial to provide statistical evidence that such a gradient exist.

The next question would be: if the mRNA stability hypothesis is true, why do the authors find the same levels of 3'UTR read coverage between both groups? Why should the mRNAs in the KO group be more susceptible to degradation, but the 3' UTR stays completely unaffected? What would make that part of the mRNAs resistant to the proposed instability?

9.

I did not find information regarding the gender of the animals used in this study.

10.

The authors show in situ hybridization for Gad45a in the hippocampus. Do they see Gadd45a in other brain regions?

11.

Can the authors test if the 12 fold overexpression of Gadd45a leads to a similar increase in protein levels?

12.

Regarding the RNA-seq data it would be interesting to show a PCA analysis to estimate the variability amongst data.

13.

Regarding the GO data: Does the p-value shown represent the value corrected for multiple testing?

14.

Loss and gain of Gadd45a function impair or enhance memory consolidation. Since in the employed loss of function model Gadd45a is lacking from early developmental stages it would be interesting to test if overexpression of Gadd45a in glutaminergic neurons also affects the 3' UTR of the identified genes.

1st Revision - authors' response

21 December 2018

Answers to referees

Referee #1:

1. In the manuscript by Rey et al. entitled "Gadd45a regulates memory consolidation via stabilization of mRNAs of synaptic plasticity proteins", the authors explore the role of the Gadd45a protein in memory through studies of an established Gadd45a knockout mouse. Through a transcriptomic analysis, the authors identify candidate targets of Gadd45a and they propose a model that Gadd45a regulates the stability of target RNAs that are critical for neuronal function. There are some aspects of the work that require more evidence to support the model. Mechanistic studies are challenging *in vivo*, but some experimental approaches could provide support for the model. The data used to argue for a change in RNA stability is not sufficient to support this model.

We thank referee #1 for the constructive input and agree on the difficulty of mechanistically proving our model *in vivo*. Nevertheless, in our revised manuscript, we present new support for the post-transcriptional nature of the action of Gadd45a, together with an extended description of why alternative explanations can be rejected (transcriptional regulation, epigenetic modifications and alternative polyadenylation usage). In addition, we identified new aspects of the modulation of memory by Gadd45a, including differences in the synaptic location of its targets and the requirement of its RNA-binding capacity to regulate memory. Together with rephrasing of statements and more detailed explanations of the experimental approach, we believe that our conclusions are strengthened.

2. The authors employ a previously generated and characterized Gadd45a knockout mouse. However, as much of the data presented relies on this mouse, the mouse should be described in some detail in the Introduction. No information is provided here about the nature of the knockout with only some minor comments about the characterized phenotypes. While characters may be limited, there is a need for at least some introduction to this mouse.

We incorporated an extended description of the *Gadd45a*-KO mouse model (Page 3, lines 87-92), with a short explanation of the nature of the generation and a more detailed description of the phenotypes previously observed.

3. Figure 1 presents the initial evidence that the Gadd45a mouse has defects in memory consolidation. The order of Figure is rather odd. The finding that Gadd45a transcript is present in the CA 1-3 regions really provides the rationale to assess hippocampal functions. Thus, current Figure 1C, which also provides evidence that the knockout mouse actually lacks Gadd45a transcript in these regions of the brain seems like the logical Figure 1A.

While we understand the reasoning, we would like to keep the order based on the following logic:

- Based on previous knowledge and open data repositories such as the Allen brain atlas (<http://mouse.brain-map.org/experiment/show/576421>), we already knew that *Gadd45a* is expressed throughout many regions in the brain, including the hippocampus. Thus, it was not the rationale for testing hippocampal-dependent learning.

- As we did not know which neuronal function(s) *Gadd45a* had, the rationale for testing all the behaviors, including those in Figure EV1, was to identify *Gadd45a*-dependent processes.
- The in situ hybridization was primarily performed to identify the neuronal subpopulation expressing *Gadd45a* mRNA (i.e., principal glutamatergic neurons), allowing then the rational application of the genetic overexpression model.

Therefore, we believe that keeping the order of the figure is not only more accurate but also makes the manuscript more logic and easy to read.

4. The image of the apparatus used for the experiment currently shown in Figure 1A is not informative and the Figure Legend does not add to the value of the image. If the authors wish to illustrate this behavioral paradigm, they could employ a schematic that illustrates the apparatus and depicts the experiment.

We changed Figure 1A accordingly. The new version is more illustrative as to the process by which the aversive learning is induced, and we also implemented a better explanation of the task in the figure legend.

5. The experiment in current Figure 1D is only mentioned in passing with no comparison of *Gadd45a* levels to *Gadd45b* or *Gadd45g*, despite the fact that these are the data presented in this figure.

We agree. Former Figure 1D-E was not important enough to be included as a main figure panel, and we therefore moved the content to Figure EV1.

6. The relative RNA comparison shown in Figure 1E is not that valuable as RNA levels do not dictate protein levels, which is really the valuable comparison. These data do raise the question of how the function of *Gadd45a* is coordinated with or complemented by the function of *Gadd45b* and/or *Gadd45g*.

We agree on the higher relevance that protein expression contains as compared to mRNA levels. However, *Gadd45a* detection at the protein level has been a remarkable challenge throughout the course of this investigation due to the lack of appropriate anti-*Gadd45a* primary antibodies. In this context, we tested 6 different antibodies (3 homemade and 3 commercial antibodies from Santa-Cruz and Abcam), constantly obtaining unspecific bands, disturbing background and/or erratic signals. Therefore, we were unable to reliably analyze the presence of *Gadd45a* at the protein level. Nonetheless, *Gadd45a* mRNA is known to be very short-lived with an approximate half-life of less than 1h [1]. This scenario suggests that *Gadd45a* mRNA is highly regulated and its presence is a good proxy of the protein level. Therefore, despite of not having data at the protein level, we believe that the mRNA level is very informative. Moreover, thanks to the fusion of hemagglutinin (HA) to *Gadd45a* in our overexpression model, we were able to partially answer some of the key questions regarding the presence of *Gadd45a* protein in different cellular compartments (see comments below).

7. These data do raise the question of how the function of *Gadd45a* is coordinated with or complemented by the function of *Gadd45b* and/or *Gadd45g*.

This is indeed an interesting topic that was incorporated in the revised discussion (Page 10, lines 405-409), which now includes also a reference to the partially redundant function of *Gadd45* proteins in DNA demethylation (Page 11, lines 474-479).

8. Figure 2 analyzes the consequences of overexpression of *Gadd45a*. The authors missed an opportunity to perform these experiments with a version of *Gadd45a* that should not binding RNA; however, the experiments presented still argue that overexpression of *Gadd45a* is sufficient to enhance performance in the memory consolidation assay employed. This finding is surprising, which is why having a control that uses a version of *Gadd45a* that cannot bind to RNA would have been ideal. In fact, amino acid substitutions that decrease the interaction of *Gadd45a* with RNA have been created and characterized (Sytnikova et al., PLOS One, 2011).

We appreciate this suggestion because it represents an elegant way to test our hypothesis regarding *Gadd45a*-mRNA interaction. Consequently, we made a new AAV-*Gadd45a* variant in which a point mutation was introduced (K45E), in order to block the RNA-binding capacity of *Gadd45a*. This experiment is now included in the manuscript in Figure 6. In agreement with our hypothesis, mice injected with the AAV-*Gadd45a*^{K45E} showed no memory improvement as compared to mice injected with wild-type AAV-*Gadd45a*, demonstrating that *Gadd45a* needs to bind RNA to regulate aversive learning. We believe that this key experiment substantially nurtures our hypothesis, corroborating the post-transcriptional potential of *Gadd45a* and defining the role of this RNA-binding protein (RBP) in the brain.

9. Figure 4 presents RNA-Seq data comparing the Control and *Gadd45a* knockout mouse. The authors argue that the transcripts altered in the *Gadd45a* samples contain an extended 3'UTR.

This is indeed our observation, but it should not be understood that *Gadd45a* regulates the length of the extended 3'UTRs. In other words, *Gadd45a* does not control the length of these extensions, but it regulates (predominantly post-transcriptionally) transcripts that contain them. We now introduced a sentence clarifying this issue (Page 6, lines 244-246). The focus on extended 3'UTR was initially based on (a) comparison of the actual RNA-seq read coverage with the standard Ensembl mm9 gene annotation, and (b) the reported non-canonical 3'UTR extensions of many hippocampal transcripts in the study of Miura *et al.* [2]. More recently, some of these 3'UTR extensions have been incorporated as alternative transcript isoforms in the updated mouse gene annotation (see <http://tinyurl.com/yddcfvjs> e.g., for *Grin2a*).

10. From the presentation of the data, it is not clear whether the authors compared this sample of transcripts to a control set from the same samples.

In the first paragraph of this section (Page 6, *Gadd45a* affects the expression of a distinct set of memory-related genes), we explain our experimental approach: “we carried out a genome-wide transcriptome analysis by mRNA-seq, using hippocampal samples of *Gadd45a*-WT (n=6) and *Gadd45a*-KO mice (n=6) collected 1 hour after the training in the PA paradigm” (Page 6, lines 227-229).

11. These extended 3'UTR could be diagnostic of the tissue used for the study rather than specific to loss of *Gadd45a*.

Indeed, the 3'UTR extensions are known to have a remarkably increased prevalence in the hippocampus [2]. In fact, we refer to this issue in the same paragraph mentioned above (i.e. “Such extended 3'UTRs have recently been reported in particular in neural tissues, including hippocampus”; Page 6, lines 243-244). Nevertheless, we believe that the concern of referee #1 might be that we misinterpreted the presence of such extensions with the action of *Gadd45a* (see previous comments). In clarification to this, we do not think that *Gadd45a* controls the existence/length of these 3'UTR extensions, but it certainly regulates the transcripts that contain them, in a way that is only explainable when taking into account mRNA stability.

12. The data presented in Figure 4E is not sufficient to argue for a difference in the 3'UTR. There could be a bigger difference in the low level reads at the start of the 3'UTR as compared to the many reads at the end of the 3'UTR.

Indeed, that is what we observe: a progressively larger difference in RNA-seq read coverage from 3' to 5', which resulted in the detection of apparent differential expression of many genes using a standard analysis approach, which considers only those NGS reads overlapping with annotated exons. Upon careful data exploration, we noticed that *Gadd45a* loss did not change transcript isoform usage but rather affected the 5'-3' coverage profiles indicative of differential RNA fragmentation or, alternatively, differential RNA protection from fragmentation during tissue lysis. We agree that the mRNA-seq data alone is only suggestive and tentative evidence and might be prone to technical biases. Therefore, we performed additional experiments for independent validation of this observation (Appendix Fig S3C-D), obtaining very similar results in an independent set of samples. Moreover, we now also performed exon-level differential expression analysis (see Table EV3), which corroborated our conclusion: During aversive learning, *Gadd45a* protects against degradation of certain mRNAs containing extended 3'UTRs.

13. Figure 5 examines the *Grin2a* transcript as a target of *Gadd45a*. The argument that a difference in random-primed cDNA as compared to oligo(dT)-primed cDNA supports a model of altered stability. A surrogate measure for transcription would be to examine pre-mRNA, which is rapidly processed and thus an acceptable measure of transcription.

We thank referee #1 for the suggestion on the pre-mRNA analysis. We performed this test using primers covering intronic regions of the two most important targets of *Gadd45a* (i.e. *Grin2a* and *Grm5*). The results indicate that *Gadd45a* does not change the levels of pre-mRNA for these transcripts (Fig EV4D), corroborating the notion that *Gadd45a* does not affect the transcriptional rate of its targets.

14. These results could be indicative of alternative cleavage/polyadenylation rather than the authors' conclusion regarding RNA stability.

Indeed, we considered this possibility, due to the fact that alternative polyadenylation (APA) contributes to the complexity of the transcriptome by generating isoforms that differ either in their coding sequence or in their 3'UTRs. This potentially regulates the function, stability, localization and translation efficiency of target RNAs. However, APA is expected to produce discrete, stepwise changes in the normalised read coverage at APA sites (Rebuttal Fig 1A), with signal drop in the opposite direction to the one we see in the RNA-seq data, namely continuous drop of read coverage from 3' to 5'. APA affects a large number of

genes, and in our RNA-seq data, APA events can be unequivocally identified (Rebuttal Fig 1B), always characterized by the stepwise decrease in read coverage towards the 3' end.

Rebuttal Figure 1. Alternative polyadenylation (APA) cannot explain the observed transcriptome changes. (A) Scheme of a theoretical transcript regulated by APA at 3 distinct polyadenylation sites. The differential usage of these sites will result in different read coverage that is characterized by stepwise decreases towards the 3' end. (B) APA events are observed in our RNA-seq data but they are not affected by the loss of *Gadd45a*.

The consistent differences in the RNA-seq coverage profiles in the WT vs. KO samples are thus not compatible with APA. We nevertheless performed formal exploration of the APA hypothesis using the published tool DaPars [3], which found no significant APA events. A paragraph dedicated to this alternative hypothesis is now included in the discussion (Page 11-12, lines 487-501). Instead, our results are rather compatible with differential RNA fragmentation (e.g. via differential nonsense-mediated decay), resulting in selective enrichment of 3' transcript ends by the TruSeq poly(A)-selection NGS protocol.

Additionally, we now analysed in more detail the exonic and intronic RNA-seq coverage of *Grin2a*, *Grin2b*, *Kcnq3* and *Grm5* (Fig EV4C) and found that the changes seen in the exonic reads are mostly absent in the intronic reads derived from immature transcripts, additionally supporting the notion of post-transcriptional effects on RNA stability and/or RNA localisation.

15. Comparing the data presented in Figure 4E and Figure 5D/E is confusing because the difference in the control levels of 3'UTR transcript levels compared to the *Gadd45a*-KO is lost in the relative mRNA levels presented. These data should be plotted on the same graph to allow comparison.

We agree with referee #1, and we accordingly changed Figure 5, embedding it into Figure 4. We also changed the variable distribution in Figure 4H-I (former Figure 5D-E) and Appendix Fig S4D-F, in order to compare between genotypes for proximal and/or distal parts of the 3'UTR, instead of comparing between these different areas for a given genotype. Finally, we also changed the color code of the graph for better comparison with the RNA-seq data. Now, white bars correspond to control groups (no PA, not included in the RNA-seq study) and color bars (black/blue) correspond to the same sample analyzed by RNA-seq. We believe that this figure rearrangement improved visualization of the effects of *Gadd45a* at the transcriptome level.

16. Figure 6 presents mechanistic insight into the difference in the level of the *Grin2a* protein product (NR2A) in the synaptosomal fraction. With these fractions in hand, the authors could address whether the levels of *Gadd45a* transcript are locally altered.

We did not investigate *Gadd45a* transcript levels in the synaptosomes, because we do not have any reason to think that *Gadd45a* mRNA is locally translated or that the effects of *Gadd45a* are mediated via its mRNA. We did though investigate the presence of *Gadd45a* protein in the synaptosomes. Of note, we do not think that *Gadd45a* protein has to reach the synaptosome to induce its effects on mRNA. Nevertheless,

in order to circumvent the problematic issue of anti-Gadd45 α antibodies (see above in the rebuttal letter), we used hippocampal samples from *Gadd45a*-overexpressing mice to detect HA (fused to Gadd45 α) by western blot.

Rebuttal Figure 2. Western blot analysis of HA protein levels (fused to Gadd45 α) in hippocampal samples from *Gadd45a*-WT, *Gadd45a*-overexpressing and *Gadd45a*^{K45E}-overexpressing mice (n=3 for all groups). Note that Gadd45 α -HA was detected in all samples from *Gadd45a*-overexpressing and *Gadd45a*^{K45E}-overexpressing mice only in the total fraction, but not in the synaptosomal fraction.

The results show that Gadd45 α -HA is not present in the synaptosomes (Rebuttal Fig 2). However, given the small proportion of post-transcriptional modifications that occur in the synaptosomes, we do not think that the absence of Gadd45 α in this cellular compartment invalidates our hypothesis. In other words, Gadd45 α can regulate the stability of memory-related mRNAs in nuclear speckles or other areas of the cytoplasm, without necessarily being present in the synaptosomes.

17. The RNA binding experiment is not very robust. With this experiment, the interaction could be indirect—is the percent input recovery really 0.25%?

We agree that the native RNA immunoprecipitation experiment indeed does not show that the interaction of Gadd45 α with mRNAs is direct. To reduce unspecific immunoprecipitation, we used high salt and denaturing conditions with 1 M urea during washing to remove any unspecific related RNA binders. In addition, the purified proteins we used for binding (Gadd45 α -FLAG and FLAG-GFP) were prepared under stringent conditions and extensively treated with benzonase, RNase A and RNase I. After binding of tissue lysates, stringent washings were also applied to decrease the percentage of bound mRNAs to get more specific binding related to Gadd45 α . Therefore, 0.25% is low but highly specific. In addition, as negative control we showed that a protein such as GFP (without reported RNA-binding capacity) binds mRNAs with a much lower affinity as compared to Gadd45 α . In summary, we think that, as a proof of concept, this experiment demonstrates that Gadd45 α specifically binds memory-related mRNAs with extended 3'UTRs, directly or indirectly. Note that Gadd45 α might act in a restricted cellular sub-population, so that the overall effect gets diluted after whole tissue lysis.

18. There are some aspects of the model that do not make sense. The Gadd45a protein is localized to nuclear speckles, which is typically a site of RNA processing. The authors do not present evidence that Gadd45a is located in the cytoplasm (or outside the nucleus). Most typically control of RNA stability occurs in the cytoplasm and the argument here might be that there are local changes to impact local NR2A levels. The authors need to address these points in a mechanistic model.

While we agree that most of the control of RNA stability occurs in the cytoplasm, it should be noted that this process is closely related to mRNA maturation, and that the proteins localized in nuclear speckles not only participate in numerous aspects of mRNA synthesis, but also influence cytoplasmic mRNA events [4]. In this context, RNA-binding proteins in nuclear speckles can leave signatures on the mRNA (e.g. deposited proteins, poly(A) tail length or modified nucleotides) that regulate its subsequent nuclear export, subcellular localization, stability and translation [4]. For instance, nuclear speckles contain the enzymatic machinery that regulates N6-methyladenosine (m⁶A) RNA modifications. Interestingly, this modification has been associated with the localization of mRNA from the translatable pool to mRNA decay sites [5,6], demonstrating the potential of nuclear RNA modification in subsequent cytoplasmic control of mRNA stability.

Nevertheless, reevaluation of the brain slides from *Gadd45a*-overexpression mice revealed a significant Gadd45 α -HA signal outside of the stratum oriens (where the nuclei of principal glutamatergic neurons are located). We changed Figure 2C to better illustrate that Gadd45 α -HA is not only restricted to the nuclear layer. These data, together with the concern of referee #3, led us to analyze by western blot the presence of HA in nuclear vs. cytoplasmic protein fractions. The results (now integrated in Fig 2D) suggest that although the Gadd45 α -HA signal is considerably stronger in the nucleus, there is also Gadd45 α -HA present in the cytoplasm. Finally, confocal microscopy analysis revealed that the Gadd45 α -HA signal extends beyond that of DAPI (Fig 6D), corroborating that Gadd45 α is not exclusively present in the nucleus but also in the cytoplasm.

We understand that this answer does not provide a mechanistic explanation of the action of Gadd45 α at the mRNA level. However, we believe that the findings of this investigation are well grounded and that the characterization of the molecular events in detail would require a different experimental approach. In this sense, *in vitro* studies using neuronal cell lines with *Gadd45a*-loss and -gain-of-function will enable to unravel molecular processes by which Gadd45 α controls mRNA stability.

19. As presented, there is not sufficient evidence to support the authors' model. With the synaptosomal fractions, the authors have the potential to directly address some of the mechanistic questions raised such as whether Gadd45 α can be detected in these fractions.

We thank referee #1 for the suggestion regarding the synaptosomal fractions. We indeed used these fractions to quantify the presence of Gadd45 α -targets, such as *Grin2a* and *Grm5*. The results (now included in Fig EV5D) show that both transcripts are significantly down-regulated in *Gadd45a*-KO samples, suggesting that the effect of Gadd45 α on RNA stability can impose consequences in the localization of the mRNAs. We believe that, despite of not having all the pieces of the model, this finding contributes to the understanding of the molecular framework in which Gadd45 α exerts its actions.

20. The Table shown in Figure 4B is very difficult to read even when viewed in a magnified PDF image. The number of significant figures shown for the fold enrichment and the p-values shown seems rather excessive.

We agree. Accordingly, in the new version of the table, less gene ontology categories are shown, and the p-values were shortened to only one decimal digit.

Referee #2:

Rey et al nicely investigated roles of Gadd45 α in learning and memory and synaptic plasticity using knockout mice and overexpression mice. The authors have shown that Gadd45 α KO mice showed impaired long-term memories but showed normal short-term memories. These KO mice also showed impaired hippocampal LTP. Conversely, Gadd45 α overexpression improved long-term memory. These observations suggest that Gadd45 α positively controls memory consolidation and LTP. Furthermore, the authors found that Gadd45 α deficiency destabilizes mRNAs for NR2A and mGluR5 that is bound with Gadd45 α , leading to reductions of these protein levels. The authors extensively performed behavioral, electrophysiological and biochemical experiments and showed important findings. The results are so clear. I have only minor concerns as follows.

1. I recommend that the authors discuss about molecular mechanism of impairments in memory consolidation and LTP by Gadd45 α KO. Although post-transcriptional roles of Gadd45 α in mRNAs for synaptic proteins are well discussed in the Discussion session, they did not well discuss about how Gadd45 α KO and overexpression impaired or improved, respectively, LTP and memory. Do the authors hypothesize that reduction of NR2A and mGluR5 is main reason leading to impaired LTP and memory?

We thank reviewer #2 for the positive feedback. Regarding the discussion on LTP, we now extended it with the following topics (Page 10-11, lines 422-436):

- Additional bibliography regarding the involvement of glutamate receptors and LTP.
- References to the new experiment in which we have detected a decreased synaptosomal expression of *Grin2a* and *Grm5* in *Gadd45a*-KO mice (Fig EV5D). Importantly, this phenotype is observed in *Gadd45a*-KO mice also under control conditions. Hence, this experiment indicates that Gadd45 α controls the presence of locally transcribed mRNAs, which are in turn important for the presence of LTP.

- Reference to the new experiment in which increased levels of *Grin2a* and *Grm5* were found in *Gadd45a*-overexpressing mice (Fig EV4E-F). Again, this was also true in control conditions, indicating that the enhanced LTP in *Gadd45a*-overexpressing mice can be explained by pre-existing increases in the expression of glutamate receptors.
- Additional information on the reduced expression of a number of important players in LTP formation (e.g., transcripts coding for ion channels that regulate neuronal excitability) in *Gadd45a*-KO mice.

We believe that the new experiments and their discussion improved the overall insights into the action of *Gadd45a* on LTP, which was an underdeveloped part of our previous version.

2. P4 (Result of Fig 1C). "The signal localization suggests that *Gadd45a* mRNA is restricted to the cell body of principal glutamatergic neurons" is an overstatement. The authors did not show pictures with higher magnification and *in situ* hybridization using other marker probes.

We agree, and thus we changed to: "The signal localization suggests that *Gadd45a* mRNA is restricted to the stratum pyramidale of the hippocampus, a layer highly populated by the cell body of principal glutamatergic neurons". We also agree that the employment of other probes in combination with higher magnification imaging will result in a more comprehensive picture of the localization of *Gadd45a* mRNA.

However, we think that protein expression is more informative, and we conducted several experiments aiming at identifying the cellular compartment(s) in which *Gadd45a* is present. Unfortunately, as explained to reviewer #1, the lack of proper anti-*Gadd45a* antibodies prevented us from reliably detecting endogenous *Gadd45a*. Nevertheless, the HA protein that is fused to *Gadd45a* in the *Gadd45a*-overexpressing model, allowed us to track the expression of *Gadd45a*-HA both by western blot and immunohistochemistry. Hence, we could demonstrate that *Gadd45a*-HA protein is expressed also in the stratum oriens and stratum radiatum, which are layers enriched in neuronal projections (Fig 2C). Moreover, nuclear/cytoplasmic fractionation revealed that *Gadd45a* is not restricted to the nucleus, but is also present in the cytoplasm (Fig 2D). This finding was corroborated by confocal microscopy on hippocampal slices from *Gadd45a*-overexpressing mice (Fig 6C). Consequently, we believe that despite of the limitations of our *in situ* hybridization experiment, we were able to investigate the subcellular localization of *Gadd45a* protein.

3. From the results of Figure 5 and sFig.9, it is a bit difficult to understand (suppose) existence forms of *Gadd45a* mRNA. I am curious for results of northern blotting using specific probes.

To our knowledge, there is only one study reporting different isoforms of *Gadd45a* [7]. According to this paper, the isoform *Gadd45a1* is derived from an alternative splicing of the *Gadd45a* mRNA by skipping the region corresponding to the exon 2 of the *Gadd45a* gene during mRNA maturation [7]. In this *in vitro* study on epithelial cells, the isoform *Gadd45a1* is able to antagonize the function of *Gadd45a* in cell cycle arrest. Therefore, we appreciate the comment of referee #2, but we believe that the study of *Gadd45a* isoforms is out of the scope of this study.

Referee #3:

Rey et al study the role of the RNA binding protein *Gadd45a* in memory function and report that loss of *Gadd45a* in mice leads to impaired synaptic plasticity and learning behavior, while its over-expression improves specific forms of memory consolidation. The authors provide evidence that *Gadd45a* is involved in the turnover of mRNA coding for synaptic proteins. This is a very interesting study on a timely topic. The link of *Gadd45a* to the stability of mRNA coding of synaptic proteins is particularly interesting, although this part is less developed than the rest of the study and - at present - remains somewhat speculative. I understand, however, that the many experiments that come to mind are beyond the scope of the manuscript and suggest that many issues can be solved via rewording and I think the study In conclusion the presented data are very intriguing and the study should be published in EMBO reports.

We appreciate the enthusiastic comment and we also acknowledge the thoughtful suggestions. We agree that some of the questions raised are certainly beyond the scope of this study. However, given the clarity and logic of the comments, we aimed at answering them experimentally. If possible within the given time frame of a revision, we aimed at obtaining new data that broaden our understanding of the model proposed here.

In the following some suggestions/questions:

1. The title somewhat overstates the data. While the authors convincingly show that Gadd45b regulates memory consolidation, the data to support the view that this finding critically depends on mRNA stabilization is interesting but would need more work to conclusively demonstrate this link. Similarly the first sentence of the discussions reads "Our study demonstrates for the first time a neural function for Gadd45a specifically in the consolidation of aversive memory and LTP, thereby contributing to the understanding of memory processing and uncovering a novel post-transcriptional mechanism involving extended 3'UTRs." Such statements should be toned down in my view.

We agree that the former title was slightly misleading since the use of the connector *via* clearly indicates a causal link that is not fully disclosed in our study. Therefore, the title has been changed to "Gadd45a regulates memory and mRNA stability of transcripts encoding synaptic plasticity proteins".

The first sentence of the discussion was changed based on the same principle, with special attention to the second part of the sentence, which is now as follows: "Additionally, it also offers a new perspective on a post-transcriptional mechanism involving extended 3'UTRs." (Page 10, lines 395-396). We feel that this sentence reflects better that the mechanism of 3'UTR-dependent control of RNA stability is not a novel mechanism, but our findings offer a new perspective, especially considering the 3'UTR extensions, which are particularly abundant in the brain. The first part of the sentence was not modified because we truly think that, despite of the previous reports on the action of Gadd45 β , this is the first study on Gadd45 α and memory. As it was also pointed out by referees #1 and #2, we toned down our conclusions and discussion.

2. I think the authors should make clear that the regulation of mRNA stability is one potential mechanisms by which Gadd45a regulated memory formation. In fact, RNA stability itself has not been tested directly nor has the concept been explained (i.e. what factors degrade RNAs, how can that be prevented, how is it initiated).

Referee #3 points to a clear shortcoming of our initially submitted manuscript, which was the absence of thorough discussion on alternative hypothesis. We updated the discussion, which now contains a section on DNA methylation (page 11, lines 472-485), and alternative polyadenylation (Page 11-12, lines 487-501), as also suggested by referee #1.

While we agree that RNA stability has not been directly tested, we think that the indirect evidences clearly suggest that Gadd45 α plays a role in this process, and that further characterization of the mechanism requires different experimental approaches (i.e. *in vitro*). Nevertheless, it should be noted that the new data obtained during the revision of this manuscript clearly excludes the possibility of Gadd45 α acting at the transcriptional level. The lack of changes in *Grin2a* and *Grm5* pre-mRNA (see comment to referee #1; Fig EV4D), and the clearly different effect that *Gadd45a* loss has on the expression of introns vs. exons of the Gadd45 α target genes (see comment to referee #1; Fig EV4C), corroborates that Gadd45 α predominantly acts at the post-transcriptional level. Moreover, we have introduced in the discussion a more detailed explanation of the nonsense-mediated mRNA decay pathway (NMD; page 10-11, lines 438-465). We believe that NMD may be the mechanism underlying the effect of Gadd45 α on mRNA stability because of its shared preference for extended 3'UTRs [8,9]. Hence, we incorporated a description of how NMD works, which factors are important and how it is regulated.

3. Moreover, Gadd45a has been described to be involved in other pathways as well, notably DNA demethylation and genomic stability. It is somewhat surprising that the authors did not test DNA-methylation. Is there a specific reason for this? At least such alternatives should be mentioned in the discussion.

We agree with referee #3 that DNA methylation should have been tested. However, as explained in the updated discussion, there are a number of reasons that led us to follow a different route.

- The function of Gadd45 proteins in DNA methylation is partially redundant and, sometimes, it is necessary to knockout all three *Gadd45* genes in order to obtain a phenotype [10].
- Analysis of *Bdnf* expression (clearly involved in memory and regulated by DNA methylation) in our RNA-seq data set revealed no significant changes between genotypes.
- Most of the differentially regulated genes shared a particular expression profile characterized by the lack of changes in read coverage towards the 3' end of the transcript, which is incompatible with transcriptional/DNA methylation changes.

- There was no change in intron read coverage in the mutants, arguing against transcriptional regulation.

These observations, together with the finding that Gadd45 α has the functions as a RBP, prompted us to explore the RNA stability hypothesis that is now compared in detail with the other potential explanations in our discussion.

4. Does Gadd45 α bind to the selected target mRNAs?

Yes, according to the experiment presented in Figure 5C of the revised manuscript, Gadd45 α binds specifically to memory-related mRNAs identified in the RNA-seq study. We do not know if this interaction is direct or indirect, but it is certainly very specific as compared to that of GFP (a protein without RNA-binding capacity). In this experiment, we also tested the presence of mitogen-activated protein kinase 6 (Map2K6), which contains a very similar 3'UTR extension as the other Gadd45 α targets, but is not regulated by Gadd45 α . The absence of Map2k6 enrichment in the Gadd45 α -bound mRNA fraction suggests that Gadd45 α does not regulate all transcripts with extended 3'UTRs but only a certain fraction. New experiments will have to be designed to clarify the mechanism by which Gadd45 α selects its targets.

5. Fig. 5D: the increase of proximal signal would imply a higher abundance of shorter transcripts, whereas longer transcripts would remain at the same concentration. The authors however conclude: "we observed that on the one hand the distal part of the *Grin2a* 3'UTR was not significantly different between genotypes and experimental groups, indicating absence of differences at the transcriptional rate" which is not evident from the data. Furthermore: according to their mRNA stability hypothesis, transcripts from KO mice should be degraded faster, thus for getting the same abundance of transcripts the transcriptional rate of KO mice would have to be actually higher. This should be discussed.

If we understand correctly, the referee's reasoning is related to alternative polyadenylation (APA), since this process is known to give rise to short and long transcripts from the same gene. We agree that an increase of proximal reads would potentially imply a higher abundance of shorter transcripts. However, this process would be seen in RNA-seq experiment as a stepwise change in read coverage around the proximal polyadenylation site, and not as what we actually observed (i.e., a very gradual change). Also, if the effect observed would be explained by different expression of short and long isoforms of the Gadd45 α target genes, the read coverage will be lower towards the 3' end of the transcript, and again, this is not the case in our samples. As it is mentioned in response to referee #1 (see Rebuttal Fig 1A), APA is indeed observed in our RNA-seq study. For instance, *Bdnf* is known to generate short and long isoforms, and in our RNA-seq data, one can see that this process creates a read coverage profile characterized by a stepwise decrease towards the 3' end (Rebuttal Fig 1B). Therefore, we believe that higher proximal signal without changes in transcription can only be explained by a post-transcriptional mechanism involving RNA stabilization.

6. Furthermore: according to their mRNA stability hypothesis, transcripts from KO mice should be degraded faster, thus for getting the same abundance of transcripts the transcriptional rate of KO mice would have to be actually higher. This should be discussed.

We propose that the transcriptional levels of the Gadd45 α target genes are unchanged in the absence of Gadd45 α , but the subsequent post-transcriptional regulation (RNA stability) leads to a Gadd45 α -dependent increase in the amount of functional (non-fragmented) transcripts. Accordingly, these transcripts show indeed increased degradation in *Gadd45a*-KO hippocampi compared to WT. Of note, this is not observed under control conditions, but only when aversive learning is induced. We understand that in the long-term, *Gadd45a*-KO mice could develop compensatory mechanisms against the decrease in RNA stability. However, it should be noted that (a) the levels of proximal and distal parts of the *Grin2a* and *Grm5* 3'UTRs are very similar between genotypes under control conditions, and (b) most of the changes were found only 1h after the PA training. In other words, the action of Gadd45 α on RNA stability is inducible, and it occurs in a restricted time window, immediately after the acquisition of the aversive memory. Therefore, any compensatory mechanism would probably require an increased time period to be detected.

7. Fig. 6A shows "activity-dependent NR2A membrane insertion (Fig 6A). Of note, this process has been previously shown to depend on local translation of NR2A, which is in turn controlled by the 3'UTR of the *Grin2a* transcript." Is Gadd45 α present in synaptosomes? As the authors mention, local translation is more likely to account for the increase of NR2A, which would imply the presence of a Gadd45 α bound and stabilized *Grin2a* transcripts at the synapse. In the discussion the authors talk about the potential of Gadd45 α mediating the "long-distance transport of RNAs to different subcellular locations (Tiedge, 2006)". Maybe in the knockout mouse that transport failed and thus the *Grin2a* never arrives at the synapse?

We understand this concern, which is also shared by referee #1, as to the expression of Gadd45 α protein in the synaptosomes. As explained above in this rebuttal letter (see answer to comment 6 by referee #1), the lack of appropriate anti-Gadd45 α primary antibodies has impeded the detection of endogenous Gadd45 α . Importantly, during mRNA maturation, RBPs can impose molecular signatures in the mRNA that can control the fate of that transcript in the cytoplasm. In this way, processes such as cellular location, translation efficiency and/or RNA stability could be regulated by Gadd45 α without being present in the synaptosome. Nevertheless, in order to circumvent the problem with anti-Gadd45 α antibodies, we used hippocampal samples from *Gadd45a*-overexpressing mice to detect HA (fused to Gadd45 α) by western blot (see Rebuttal Fig 2). The results show that Gadd45 α -HA is not expressed in the synaptosomes, but as discussed above, we do not think that this finding invalidates the proposed mechanism of action.

8. In the discussion the authors talk about the potential of Gadd45a mediating the "long-distance transport of RNAs to different subcellular locations (Tiedge, 2006)". Maybe in the knockout mouse that transport failed and thus the *Grin2a* never arrives at the synapse?

We measured the presence of *Grin2a* and *Grm5* in the synaptosomal fractions, obtaining a significant decrease for both transcripts in *Gadd45a*-KO hippocampi (Fig EV5D). Thus, we can conclude that loss of Gadd45 α has an impact on the cellular localization of memory-related mRNAs, a process that is probably caused by changes in mRNA stability.

9. Fig 4D&E establish very clearly that in KO mice much less *Grin2a* transcripts contain exons than in the WT. But in figure 5A the authors see the same quantity of *Grin2a* RNA between both groups when using primers against the very same exons that they have shown to be way less abundant (p.7: "To validate these findings, we performed qPCR for *Grin2a* (Taqman assay covering exons 3-4) on RNA extracted from hippocampus after 1 hour following PA training, using cDNA reverse-transcribed either with random primers or with oligo(dT) primers (priming at the 3' end of polyadenylated RNAs)"). This should be clarified.

This is an important point that has not been properly explained in the initial version of the manuscript. First of all, Figure 5 is now integrated into Figure 4. Additionally, the group distribution in Figures 4H-I and Appendix Figure S4D-F was changed for a better comparison between genotypes in proximal or distal parts of the 3'UTR. We understand that current Figure 4F seems certainly contradictory when comparing with current Figure 4D and 4G. Importantly, it should be noted that for the RNA-seq study and the validation with oligo(dT)-primed cDNA, only polyadenylated RNAs are detected. On the other hand, random-primed cDNA is obtained from total RNA independently of the polyadenylation. In this sense, fragmented RNA will be reverse-transcribed into cDNA when using random primers, but not with oligo(dT) primers. As seen in the new Appendix Fig S2, in a context of RNA instability and lack of transcriptional changes, detection of differences is only possible when looking at polyadenylated RNA. On the contrary, the use of random-primed cDNA prevents the detection of genotype differences, because the cDNA derived from fragmented RNA will hinder potential effects on RNA stability. We think that the conceptual distinction between random- and oligo(dT)-priming strategies is key in the understanding of our work, and consequently, we included a paragraph dedicated to this issue in the discussion (Page 11, lines 465-470).

10. p. 6: "As a result, the read coverage differences of *Grin2a* exons between *Gadd45a*-WT and *Gadd45a*-KO are clearly observable throughout the protein coding sequence, at the beginning of the 3'UTR but not at the 3' end of the extended (non-canonical) 3'UTR. This observation suggested an alternative interpretation of the mRNA-seq results, namely a *Gadd45a*-dependent post-transcriptional mechanism that does not affect the total amount of initially transcribed mRNA, but rather mRNA stability." The authors neither provide literature that would point in the mRNA stability direction, nor do they exclude other possibilities by reasoning and/or experiments.

As already mentioned in the answers to comments 2 and 3, we now updated the discussion with a deeper and more detailed explanation of other possibilities (alternative polyadenylation, DNA methylation). In addition, we extended the description of the mRNA decay pathway that might be controlled by Gadd45 α (non-sense mediated decay, NMD). We think that the revised manuscript is more comprehensive and objective towards the intrinsic value of our findings.

11. The authors talk about a significant exon gradient (Fig. 4E description "Detailed view of the extended 3'UTR of *Grin2a* showing a significant 5'<3' gradient in read coverage. Note that the read coverage is similar between genotypes at the actual hippocampal 3'UTR end (lower panel at right side), but differs significantly in the canonical annotated 3'UTR and over all exons (e.g. exons 4-8)."), but do not provide statistical prove for that significance. R packs like DEXseq could do such analyses. Moreover inspection by eye does not reveal a clear gradient between 5' to 3' but rather a global dampening of exon usage.

We now performed differential expression analysis on the exon level using the Bioconductor package DESeq2, and the results (shown in Appendix Fig S4C and Table EV3) clearly demonstrate that there is a reduction of RNA-seq reads over most of the annotated exons. However, this effect is lost in the distal ends of the extended 3' UTRs in line with the visual inspection of the normalised RNA-seq read tracks.

12. Sentences like "However, the consistently reduced steepness of this coverage gradient in KO as compared to WT is indicative of increased transcript instability associated with Gadd45a deficiency." (p.6) are not well grounded. If the mRNA stability hypothesis was true, then in line with the result that the 3' UTR read coverage is the same between WT and KO, the only possibility of mRNA decay in Gadd45a-KO mice would be an RNase activity from 5'→3' (which is one of the known mechanisms for mRNA decay. As this would happen at different time points from cell to cell and transcript to transcript one would expect indeed a gradient in the exonal coverage: more transcripts would be captured with at least a few exons fully covered towards the 3' end, whereas almost no transcript would be seen with reads over the first exons (since they would then be the first ones getting decayed). Thus, it would be crucial to provide statistical evidence that such a gradient exist.

As explained in our previous comment, DESeq2 analysis at the exon level resulted in a very similar difference in read coverage between genotypes, across all the exons (Table EV3). This means that 5' exons are equally down-regulated as compared to 3' exons in KO as compared with WT. We understand the logic presented by referee #3, but it is important to note that NMD, as a candidate mechanism driving Gadd45a-mediated RNA stabilization, is known to start by endonucleolytic cleavage at premature stop codons. In extended 3'UTRs, this can happen at the canonical polyadenylation site (proximal 3'UTR), due to the distance to the actual polyadenylation site (e.g., 10 kb downstream the canonical polyadenylation site of the *Grin2a* transcript). This leaves the rest of the 3'UTR exposed to 5' exonucleases, creating indeed a 5'<3' gradient in the read coverage (Fig 4E). However, the 5' fragments that NMD generates and that contain exons are susceptible to both 5' and 3' exonucleases (because of the lack of a protective poly(A) tail at the 3' end), finally leading to a very similar degree of degradation from both ends.

13. The next question would be: if the mRNA stability hypothesis is true, why do the authors find the same levels of 3'UTR read coverage between both groups? Why should the mRNAs in the KO group be more susceptible to degradation, but the 3' UTR stays completely unaffected? What would make that part of the mRNAs resistant to the proposed instability?

First, we only find the same levels of 3'UTR coverage at the very end of the 3'UTR. The difference progressively increases towards the 5' end of the 3'UTR. In our opinion, what explains the same read coverage at the very end of the 3'UTR is a similar/same rate of transcription, which is now confirmed by the exon vs. intron analysis and the pre-mRNA levels (Fig EV4C-D, respectively).

The mRNAs in *Gadd45a*-KO mice could be more susceptible for degradation because of a number of reasons that are tentative at the moment. Therefore, we decided not to include them in the discussion. Further characterization of the mechanism will probably require the implementation of a different experimental approach (i.e. *in vitro*). But just to mention a few reasons:

- Given that Gadd45a binds RNA and it is also known to bind TET dioxygenases, Gadd45a might change the methylation status of these transcripts, a mark that is known to control RNA stability.
- AU-rich elements (AREs) are also known to be important in controlling RNA stability. A potential mechanism of action for Gadd45a could be the protection of certain mRNAs by preventing the action of the enzymatic machinery responsible for recognition and processing of AREs.

14. I did not find information regarding the gender of the animals used in this study. Male mice were used in this study (Page 13, line 549).

15. The authors show *in situ* hybridization for Gadd45a in the hippocampus. Do they see Gadd45a in other brain regions?

Gadd45a is also expressed in other brain regions, especially in the olfactory areas and the cerebellum. Given the goal-directed nature of the *in situ* hybridization presented here (i.e. focus on hippocampal expression), we did not look at rostral and caudal regions of the brain, where the above mentioned areas are located.

16. Can the authors test if the 12 fold overexpression of Gadd45a leads to a similar increase in protein levels?

As explained above, we could not reliably measure endogenous Gadd45 α level because of the lack of appropriate anti-Gadd45 α antibodies.

17. Regarding the RNA-seq data it would be interesting to show a PCA analysis to estimate the variability amongst data.

We now incorporated a PCA analysis (Appendix Fig S4B), which shows a relatively low variability between samples, especially in the *Gadd45a*-WT group.

18. Regarding the GO data: Does the p-value shown represent the value corrected for multiple testing?

In Table EV2, all 3 common multiple-testing corrections computed by DAVID are shown (Bonferroni, Benjamini, and FDR), in addition to the raw p-value from the gene set enrichment analysis. In Figure 4B, excerpts of only the raw p-value are given due to space constraints.

19. Loss and gain of Gadd45a function impair or enhance memory consolidation. Since in the employed loss of function model Gadd45a is lacking from early developmental stages it would be interesting to test if overexpression of Gadd45a in glutamatergic neurons also affects the 3' UTR of the identified genes.

We now investigated the expression of *Grin2a* and *Grm5* in hippocampal samples from *Gadd45a*-overexpressing mice (Fig EV4E-F), revealing a significant increase of both transcripts as compared to *Gadd45a*-WT samples. Of note, the increase was found in both control and conditioned animals, which suggests that 5-weeks of *Gadd45a*-overexpression (time between intracranial AAV-delivery and sample collection) is sufficient to change the expression of these memory-related genes. Surprisingly, the increase was also detected when measuring random-primed cDNA (although not statistically significant for *Grin2a*), which indicates that *Gadd45a*-overexpression can also have transcriptional effects. This is not well correlated with the identified action of endogenous Gadd45a (mostly post-transcriptional). However, it is not entirely unexpected, given the high number of Gadd45a interaction partners that can potentially regulate transcription (mitogen-activated protein kinases, transcription factors, etc.), and the extent of the overexpression (12-16 fold at the mRNA level). Nevertheless, this finding in *Gadd45a*-overexpressing animals, together with the decrease in synaptosomal levels in *Gadd45a*-KO samples (Fig EV5D), also provides the molecular framework understanding the underpinnings of the decreased and enhanced LTP observed, respectively.

Bibliography of rebuttal letter

1. Moskalev AA, Smit-McBride Z, Shaposhnikov MV, Plyusnina EN, Zhavoronkov A, Budovsky A, Tacutu R, Fraifeld VE (2012) Gadd45 proteins: relevance to aging, longevity and age-related pathologies. *Ageing research reviews* **11**: 51-66
2. Miura P, Shenker S, Andreu-Agullo C, Westholm JO, Lai EC (2013) Widespread and extensive lengthening of 3' UTRs in the mammalian brain. *Genome research* **23**: 812-825
3. Xia Z, Donehower LA, Cooper TA, Neilson JR, Wheeler DA, Wagner EJ, Li W (2014) Dynamic analyses of alternative polyadenylation from RNA-seq reveal a 3'-UTR landscape across seven tumour types. *Nature communications* **5**: 5274
4. Galganski L, Urbanek MO, Krzyzosiak WJ (2017) Nuclear speckles: molecular organization, biological function and role in disease. *Nucleic acids research* **45**: 10350-10368
5. Wang X, Lu Z, Gomez A, Hon GC, Yue Y, Han D, Fu Y, Parisien M, Dai Q, Jia G, *et al.* (2014) N6-methyladenosine-dependent regulation of messenger RNA stability. *Nature* **505**: 117-120
6. Fu Y, Dominissini D, Rechavi G, He C (2014) Gene expression regulation mediated through reversible m(6)A RNA methylation. *Nature reviews. Genetics* **15**: 293-306
7. Zhang Y, Beezhold K, Castranova V, Shi X, Chen F (2009) Characterization of an alternatively spliced GADD45alpha, GADD45alpha1 isoform, in arsenic-treated epithelial cells. *Molecular carcinogenesis* **48**: 454-464
8. Buhler M, Steiner S, Mohn F, Paillusson A, Muhlemann O (2006) EJC-independent degradation of nonsense immunoglobulin-[mu] mRNA depends on 3[prime] UTR length. *Nat Struct Mol Biol* **13**: 462-464
9. Kertesz S, Kerényi Z, Merai Z, Bartos I, Palfy T, Barta E, Silhavy D (2006) Both introns and long 3'-UTRs operate as cis-acting elements to trigger nonsense-mediated decay in plants. *Nucleic acids research* **34**: 6147-6157
10. Rai K, Huggins IJ, James SR, Karpf AR, Jones DA, Cairns BR (2008) DNA demethylation in zebrafish involves the coupling of a deaminase, a glycosylase, and gadd45. *Cell* **135**: 1201-1212

Thank you for the submission of your revised manuscript to our editorial offices. We have now received the reports from the three referees that were asked to re-evaluate your study, you will find below. As you will see, the referees now support the publication of your manuscript in EMBO reports. However, referee #1 has some remaining concerns or further suggestions we ask you to address in a final revised version of your manuscript.

Further, I have these editorial requests:

- Please revise the title of the manuscript (see comment of referee #1). Please be sure that the new title has not more than 100 characters (including spaces).
- Please provide the abstract written in present tense.
- Please add a paragraph describing the statistical methods used throughout the manuscript to the methods section.
- Please add scale bars to panels 3A and EV3E.
- Please provide single editable TIFF or EPS-formatted figure files (for main figures and EV figures) in high resolution.
- Please add a paragraph describing the author contributions to the manuscript (before the acknowledgements).
- Per journal policy, we do not allow 'data not shown' (see page 21 of your manuscript). All data referred to in the paper should be displayed in the main or Expanded View figures, or the Appendix. Thus, please add these data (or change the text accordingly, if these data are not important). See: <http://embor.embopress.org/authorguide#unpublisheddata>
- The Appendix figures are not called out sequentially in the manuscript. Please change their order and nomenclature in the Appendix file to fix this.
- It seems figure 6E is presently not called out. Please add a callout to the manuscript text.
- As they are significantly cropped, could you provide the source data for the entire Western blots shown in the manuscript (including the EV figures)? The source data will be published in separate source data files online along with the accepted manuscript and will be linked to the relevant figures. Please submit scans of entire gels or blots together with the revised manuscript. Please include size markers for scans of entire gels, label the scans with figure and panel number, and send one PDF file per figure.
- Some of the WB panels are over-contrasted (e.g. in Fig 2D, 6C and EV5B/C). Please show the WBs with panels with equal contrast, and as unmodified as possible.
- EV tables 1 and 2 are too large to be displayed online. Please upload these as datasets, and change their callouts accordingly (Dataset EVx). Please upload these files as original excel sheet, with the legend on the first tab, and remove the respective legends from the Appendix.
- Please add tables EV3 and EV4 to the Appendix, above their legends that are already in the Appendix. Please call these Appendix Table S1 and S2, and change their callouts in the manuscript text, and add them to the Appendix TOC.
- In the Appendix file, please move the legends below the respective Figures. This is much easier for the readers to comprehend.
- Please find attached a word file of the manuscript text (provided by our publisher) with changes we ask you to include in your final manuscript text, and some queries, we ask you to address. Please provide your

final manuscript file with track changes, in order that we can see the modifications done.

- a Microsoft Word file (.doc) of the final revised manuscript text.
- editable TIFF or EPS-formatted single figure files (main figures and EV figures) in high resolution.
- The revised Appendix.
- The source data for the Western blots.

REFEREE REPORTS

Referee #1:

In the revised manuscript by Rey et al. now entitled "Gadd45a regulates memory consolidation and stability of transcripts encoding synaptic plasticity proteins", the authors explore the role of the Gadd45a protein in memory through studies of an established Gadd45a knockout mouse as well as an overexpression mouse that they generate. Through a transcriptomic analysis, the authors identify candidate targets of Gadd45a and they propose a model that Gadd45a regulates the stability of target RNAs that are critical for neuronal function. The authors have gone to great lengths to address a previous round of critiques/suggestions that have improved the study. There are some points that could still be improved.

The authors still employ the term transcript stability in the title. They should really use the term post-transcriptional regulation. While they have reasonable arguments that transcription of specific mRNAs examined is not altered, they never directly examine transcript stability so having this term in the title seems like an overreach. They can certainly suggest that the most likely model is that stability is altered, but they have no idea of mechanism and some of their data are a bit confusing (points raised by Reviewer 3).

Specific Comments:

The authors rebut this previous point "The finding that Gadd45a transcript is present in the CA 1-3 regions really provides the rationale to assess hippocampal functions. Thus, current Figure 1C, which also provides evidence that the knockout mouse actually lacks Gadd45a transcript in these regions of the brain seems like the logical Figure 1A." They provide a reasonable argument but the real problem is that the Introduction provides no rationale to study the function of Gadd45a in the brain or behavior. The authors state in the rebuttal that Gadd45a is already known to be expressed in the brain, but that is not mentioned. The authors simply need to provide some rationale beyond "Despite of the extensive bibliography in the field of cancer research [13,14], a role for Gadd45 α in the central nervous system has so far not been reported." If anything this sentence makes the study look a bit misguided because there does not seem to be rationale to pursue the line of investigation described here. Presumably, there is rationale, but the authors need to convey this information.

The authors provided additional mechanistic support for their model by including overexpression of a Gadd45 α variant that has been characterized to decrease RNA binding. They should be careful with the language because what their data show is that the RNA binding function of Gadd45 α is required for the increase in memory. Ideally, they would have used this protein variant in the RNA binding experiments shown in Figure 5C. This control would have made the low enrichment binding data more compelling (assuming no binding with this variant) and linked the RNAs examined for binding to the memory phenotype described in Figure 6.

The authors include a long discussion of NMD in the Discussion. This description is rather rambling and hard to follow. In contrast, the authors do not mention miRNA-mediated regulation, which could control mRNA stability, particularly for mRNAs with long 3'UTRs. The authors have no idea of mechanism. Indeed, they do not formally know that stability is affected so they might want to provide brief suggestions and be more inclusive than in the current Discussion. It seems as if some of the studies added to the current version of the manuscript could implicate Gadd45 α in proper trafficking of the synaptic plasticity transcripts, which could in turn influence stability (via altered interaction with miRNAs or local

translation). Thus, the authors need to work on a more inclusive and logical flow to the Discussion.

Minor points:

The authors should clearly introduce the proteins, which seem to be alpha, beta, gamma and are introduced in a reasonable manner and the corresponding genes, which seem to be a,b,c. This is never clearly stated so it sometimes appears that the symbol font was just lost.

Referee #2:

I have no further concern.

Referee #3:

The authors addressed all of my previous concerns.

2nd Revision - authors' response

22 February 2019

Referee #1:

In the revised manuscript by Rey et al. now entitled "Gadd45a regulates memory consolidation and stability of transcripts encoding synaptic plasticity proteins", the authors explore the role of the Gadd45a protein in memory through studies of an established Gadd45a knockout mouse as well as an overexpression mouse that they generate. Through a transcriptomic analysis, the authors identify candidate targets of Gadd45a and they propose a model that Gadd45a regulates the stability of target RNAs that are critical for neuronal function. The authors have gone to great lengths to address a previous round of critiques/suggestions that have improved the study. There are some points that could still be improved.

The authors still employ the term transcript stability in the title. They should really use the term post-transcriptional regulation. While they have reasonable arguments that transcription of specific mRNAs examined is not altered, they never directly examine transcript stability so having this term in the title seems like an overreach. They can certainly suggest that the most likely model is that stability is altered, but they have no idea of mechanism and some of their data are a bit confusing (points raised by Reviewer 3).

We have changed the title to: "Gadd45 α modulates aversive learning through post-transcriptional regulation of memory-related mRNAs". We agree with referee 1 that the term post-transcriptional regulation describes more accurately the molecular process that is regulated by Gadd45 α . With this new title, we avoid using the term mRNA stability since indeed our evidences are not completely conclusive in this regard.

Specific Comments:

The authors rebut this previous point "The finding that Gadd45a transcript is present in the CA 1-3 regions really provides the rationale to assess hippocampal functions. Thus, current Figure 1C, which also provides evidence that the knockout mouse actually lacks Gadd45a transcript in these regions of the brain seems like the logical Figure 1A." They provide a reasonable argument but the real problem is that the Introduction provides no rationale to study the function of Gadd45a in the brain or behavior. The authors state in the rebuttal that Gadd45a is already known to be expressed in the brain, but that is not mentioned. The authors simply need to provide some rationale beyond "Despite of the extensive bibliography in the field of cancer research [13,14], a role for Gadd45 α in the central nervous system has so far not been reported." If anything this sentence makes the study look a bit misguided because there does not seem to be rationale to pursue the line of investigation described here. Presumably, there is rationale, but the authors need to convey this information.

We incorporated a sentence in the introduction referring to the expression of Gadd45 α in brain regions controlling cognitive functions (pages 1-2, lines 85-87). We believe that the introduction already provides the rationale for the study:

1.- Memory can be regulated post-transcriptionally by RBPs that interact with 3'UTRs,

- 2.- Gadd45 α has been recently described as an RBP and is present in neurons, where 3'UTRs are particularly extended,
- 3.- Previous studies on the neuronal role of other Gadd45 proteins are inconsistent,
- 4.- There is a gap of knowledge regarding the action of Gadd45 α in the brain, and its possible role as a 3'UTR-interacting RBP.

The authors provided additional mechanistic support for their model by including overexpression of a Gadd45 α variant that has been characterized to decrease RNA binding. They should be careful with the language because what their data show is that the RNA binding function of Gadd45 α is required for the increase in memory.

We have changed the heading of the last paragraph of the result section (page 9, lines 375-376), so that the conclusion regarding the importance of the RNA-binding capacity of Gadd45 α is circumscribed only to the Gadd45 α -overexpression model (increase in memory).

Ideally, they would have used this protein variant in the RNA binding experiments shown in Figure 5C. This control would have made the low enrichment binding data more compelling (assuming no binding with this variant) and linked the RNAs examined for binding to the memory phenotype described in Figure 6.

We agree with referee 1 on the convenience of using this variant of Gadd45 α for the RNA-binding experiments. However, given the ample evidence and the novelty of the findings provided in this manuscript, we feel that an in-depth analysis of the nature of the Gadd45 α -mRNA interaction would be better approached in a new and independent project. Therefore, we think that this experiment is out of the scope of this study, but it will surely be part of upcoming investigations.

The authors include a long discussion of NMD in the Discussion. This description is rather rambling and hard to follow.

The discussion of NMD was prolonged during the first revision of the manuscript due to concerns raised by referee 3 regarding the RNA stability hypothesis: "*what factors degrade RNAs, how can that be prevented, how is it initiated*". We agree with referee 1 that in its current form, this part of the discussion is poorly structured and unnecessarily detailed. Thus, we have removed some parts (page 11, lines 472-478), in order to improve the logical flow of reasoning, which goes as follows:

- 1.- NMD affects more frequently mRNAs with long 3'UTRs,
- 2.- NMD is regulated by numerous RBPs,
- 3.- NMD crucial factors have been previously linked with memory deficits,
- 4.- The molecular outcome expected when NMD is disrupted, correlates well with our observations in Gadd45 α -KO mice.

In contrast, the authors do not mention miRNA-mediated regulation, which could control mRNA stability, particularly for mRNAs with long 3'UTRs. The authors have no idea of mechanism. Indeed, they do not formally know that stability is affected so they might want to provide brief suggestions and be more inclusive than in the current Discussion.

We agree on the necessity of expanding the discussion to alternative explanations, amongst which, miRNA-dependent regulation deserves special attention. Therefore, we have included a new paragraph in the discussion that briefly explains the current knowledge on the axis memory-miRNAs-Gadd45 α (page 11, lines 455-466).

It seems as if some of the studies added to the current version of the manuscript could implicate Gadd45 α in proper trafficking of the synaptic plasticity transcripts, which could in turn influence stability (via altered interaction with miRNAs or local translation). Thus, the authors need to work on a more inclusive and logical flow to the Discussion.

Indeed, the data shown in Figure EV5D suggests that Gadd45 α controls trafficking of memory-related genes, and the discussion was changed accordingly during the first round of revision. We agree with referee 1 that in this context, local differences on miRNAs and translation machinery must be mentioned as a potential mechanism explaining our results. Consequently, we included two sentences in the discussion (page 13, lines 550-552 and 558-560) that make it more inclusive and open to alternative explanations.

Minor points:

The authors should clearly introduce the proteins, which seem to be alpha, beta, gamma and are introduced in a reasonable manner and the corresponding genes, which seem to be a,b,c. This is never clearly stated so it sometimes appears that the symbol font was just lost.

A clarifying sentence in this regard was implemented in the abstract (page 2, line 49). In addition, all Gadd45 terms were double-checked for appropriate symbol allocation.

Corresponding Author Name: Prof. Beat Lutz

Journal Submitted to: EMBO reports

Manuscript Number: EMBOR-2018-46022